# What Has Been Overlooked in Contrastive Source-Free Domain Adaptation: Leveraging Source-Informed Latent Augmentation within Neighborhood Context

**Jing Wang**                                                         *jing.wang@mech.ubc.ca*
*Department of Mechanical Engineering*
*University of British Columbia*

**Wonho Bae**                                                             *whbae@cs.ubc.ca*
*Department of Computer Science*
*University of British Columbia*

**Jiahong Chen**                                                     *jiahong.chen@ieee.org*
*Department of Mechanical Engineering*
*University of British Columbia*

**Kuangen Zhang**                                         *kuangen.zhang@alumni.ubc.ca*
*Department of Mechanical Engineering*
*University of British Columbia*

**Leonid Sigal**                                                            *lsigal@cs.ubc.ca*
*Department of Computer Science*
*University of British Columbia*

**Clarence W. de Silva**                                             *desilva@mech.ubc.ca*
*Department of Mechanical Engineering*
*University of British Columbia*

**Reviewed on OpenReview:** *https://openreview.net/forum?id=iulMde3dP1*

## Abstract

Source-free domain adaptation (SFDA) involves adapting a model originally trained using a labeled dataset (*source domain*) to perform effectively on an unlabeled dataset (*target domain*) without relying on any source data during adaptation. This adaptation is especially crucial when significant disparities in data distributions exist between the two domains and when there are privacy concerns regarding the source model's training data. The absence of access to source data during adaptation makes it challenging to analytically estimate the domain gap. To tackle this issue, various techniques have been proposed, such as unsupervised clustering, contrastive learning, and continual learning. In this paper, we first conduct an extensive theoretical analysis of SFDA based on contrastive learning, primarily because it has demonstrated superior performance compared to other techniques. Motivated by the obtained insights, we then introduce a straightforward yet highly effective latent augmentation method tailored for contrastive SFDA. This augmentation method leverages the dispersion of latent features within the neighborhood of the query sample, guided by the source pre-trained model, to enhance the informativeness of positive keys. Our approach, based on a single InfoNCE-based contrastive loss, outperforms state-of-the-art SFDA methods on widely recognized benchmark datasets. The code for our implementation: `https://github.com/JingWang18/SiLAN`.

# 1   Introduction

Supervised learning has proven successful in mimicking human behaviors in situations such as manipulation Mnih et al. (2015), recognition Russakovsky et al. (2015), and understanding Fawzi et al. (2022), primarily due to its accessibility to vast amounts of labeled data. However, this success hinges upon the assumption that both the training and the test data originate from the same underlying probability distribution, which often does not hold in various real-world scenarios. Consequently, model performance tends to deteriorate when applied to novel (*target*) data domains – such as real-world images – whose underlying data distribution markedly deviates from that of the training (*source*) domain (e.g., computer-generated images). This divergence in data distribution is commonly referred to as *domain shift* Ben-David et al. (2010). Domain adaptation (DA) tackles the performance degradation that stems from the domain shift, by acquiring representations that remain invariant to such shifts Ganin & Lempitsky (2015).

Lately, a significant and practical challenge has emerged, known as *source-free domain adaptation* (SFDA). This challenge revolves around the goal of developing the domain-invariant representation without using any labeled source data during the adaptation. This shift in focus is motivated by real concerns related to training data privacy and intellectual property Liang et al. (2020); Yang et al. (2021b); Huang et al. (2021a); Zhang et al. (2022); Yang et al. (2022). The inability to access the source domain during adaptation adds complexity to the task of estimating the degree of domain shift, making it challenging to learn a shared representation that bridges divergent domains. In response to this, the concept of contrastive learning has attracted significant attention. In this approach, discriminative clustering plays a crucial role in defining the decision boundaries that guide the classification efforts, by utilizing these clustered groups. However, in scenarios where labels are unavailable, the decision boundaries inferred from these clusters may not consistently align with the actual classification boundaries based on the target labels.

Existing contrastive SFDA methods can be categorized into two types of approaches, based on how they generate positive keys for contrastive learning, whether neighborhood-based or augmentation-based. The neighborhood-based approaches rely on transferring neighborhood information from the source to the target by initializing the model for target adaptation with a model pre-trained on the source domain Yang et al. (2022). However, as the adaptation progresses and the model undergoes unsupervised clustering, the significance of this source domain neighborhood information diminishes. Conversely, the augmentation-based approach establishes a source-like set with a distribution aligned with the source domain during pre-training for generating pseudo-labels. Subsequently, it employs contrastive learning with data augmentation to facilitate discriminative clustering during target adaptation Zhang et al. (2022). However, this method encounters two significant problems. Firstly, like pseudo-labeling-based SFDA, its performance can be negatively affected by the presence of noisy labels Liang et al. (2021a). Secondly, achieving an optimal tradeoff between distribution alignment and contrastive clustering is challenging, especially when the source data is inaccessible during target adaptation.

The primary goal of our study is to conduct a thorough analysis of how the design of positive keys could impact target domain classification performance. Moreover, we seek to explore how insights obtained from this analysis can be utilized to enhance contrastive SFDA frameworks. This understanding is grounded in empirical observations within the latent feature space, as illustrated in Figure 1. In this figure, we can observe that target features extracted from the source pre-trained encoder exhibit significant dispersion because of domain shift, leading to increased target classification errors. Despite this feature dispersion, it is worth noting that nearby target features often share the same ground truth labels. These two observations are elaborated upon as follows:

- **Observations on Target Feature Dispersion**: A domain shift induces a significant dispersion of target features, as extracted from the source pre-trained encoder (as clear when comparing the t-SNE plots for source data features and target data features). This dispersion reduces the discriminative quality of the associated features, consequently making it difficult to effectively classify samples within the target domain.

- **Observations on Neighborhood Informativeness**: Despite the reduced discriminability, which leads to the lack of well-defined target feature clusters extracted from the source pre-trained encoder, neigh-

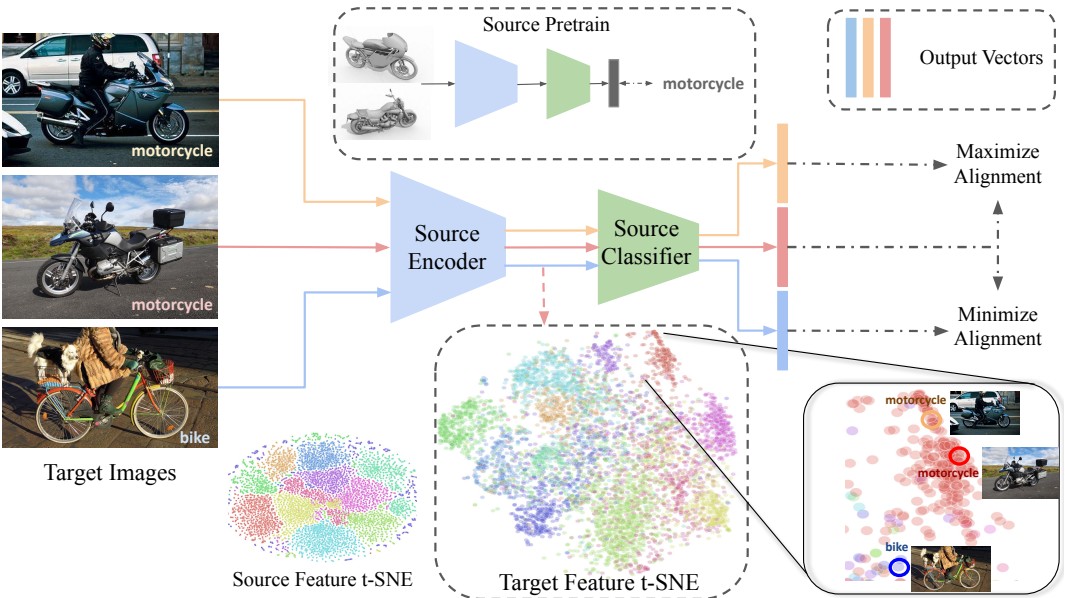

Figure 1: t-SNE visualization of target features extracted by the source pre-trained encoder, revealing significant feature dispersion due to domain shift. However, nearby target samples still tend to share similar ground truths, motivating our in-depth exploration of contrastive clustering in SFDA.

boring target features still tend to belong to the same ground truth class. This suggests that valuable information, *w.r.t* target ground truth, likely exists within the local vicinity of target features.

Motivated by these observations, we conduct a comprehensive analysis of the underlying principles of contrastive SFDA and explore aspects that have been overlooked in the existing contrastive SFDA frameworks. To be specific, we identify three overlooked factors: 1) standard data augmentation techniques might not effectively reduce the likelihood of the model misclassifying positive transformations; 2) increasing the number of nearest neighbors, i.e., larger $k$, results in smoother predictions but also leads to a greater overlap of logit clusters; and 3) effective utilization of the source pre-trained model to leverage neighborhood label consistency for enhancing the informativeness of positive key generation has not yet been explored. Subsequently, we introduce our Source-informed Latent Augmented Neighborhood (SiLAN) method, built upon the acquired theoretical insights. This involves applying Gaussian noise to the latent features of the neighborhood centroid of a target query sample, mirroring the source model's standard deviation of latent features from the neighboring target samples, to enhance positive key generation. This latent augmentation is consistently applied during the positive key generation process for each target query sample, combining the benefits of both neighborhood exploration and data augmentation. Aligning the standard deviation of the random noise with that determined by the source pre-trained model allows contrastive clustering to effectively leverage the dispersion of the neighbors' features, which is often regarded as potentially detrimental to the model's discriminability. Our empirical findings demonstrate that optimizing an InfoNCE-based contrastive loss Oord et al. (2018); Chen et al. (2020) alone, combined with our SiLAN augmentation method, yields state-of-the-art performance across a range of benchmark SFDA datasets.

In summary, our paper's contributions are outlined as follows:

- We hypothesize that domain shift causes significant dispersion in target features yet nearby points still tend to share similar labels, explaining the success of contrastive clustering in SFDA.

- Our theoretical analysis of contrastive SFDA reveals that three often-overlooked factors, associated with the aforementioned hypotheses, have significant implications for target classification performance.

- To address these three issues, we introduce SiLAN, a simple yet effective latent augmentation technique explicitly designed to improve contrastive SFDA.

- Experimental results support our theoretical findings, demonstrating that InfoNCE, when augmented with SiLAN, achieves state-of-the-art performance in SFDA.

## 2 Related Work

### 2.1 Source-Free Domain Adaptation

The existing methods for SFDA can be grouped into two categories: contrastive and non-contrastive. Non-contrastive methods rely on extra guidance, such as pseudo labels Liang et al. (2020; 2021b) or samples generated through adversarial learning Li et al. (2020), from the source pre-trained model during adaptation. However, this additional guidance can potentially have detrimental effects on target classification performance Li et al. (2020). In this paper, we focus on contrastive SFDA methods, which have shown better performance. Attracting and Dispersing (AaD) Yang et al. (2022) identifies nearby samples to a target query using neighborhood searching and directly uses their latent features as positive keys. Historical Contrastive Learning (HCL) Huang et al. (2021a) introduces an instance-wise loss to enhance data-augmentation-based contrastive learning for target adaptation. Divide and Contrast (DaC) Zhang et al. (2022) investigates contrastive clustering using augmented inputs, effectively applying it to SFDA. However, existing contrastive SFDA methods overlook the alignment of different domains during target adaptation. In contrast, our approach incorporates domain alignment into the contrastive clustering process, which is achieved while optimizing a single contrastive loss.

### 2.2 Latent Augmentation

Data augmentation can change the semantics of input samples by causing major shifts in the latent space Upchurch et al. (2017). These techniques also vary based on data modality. Conversely, augmenting in the latent space Cheung & Yeung (2020) offers diverse transformations with a lower risk of altering the semantics of queries and is not dependent on the data modality. Common latent augmentation techniques include interpolation, extrapolation, random translation, and adding Gaussian noise DeVries & Taylor (2017), which have been shown effectiveness in various fields such as computer vision Liu et al. (2018); Stutz et al. (2019), natural language processing Kumar et al. (2019), and graph representation learning Cheng et al. (2022). In this paper, we investigate using latent augmentation to address challenges in contrastive SFDA. Specifically, we tailor the variance of the random noise used in latent augmentation according to the feature cluster variances from the source pre-trained model. This key technical distinction differentiates our method from the existing latent augmentation approaches.

### 2.3 Contrastive Learning

Contrastive learning is an effective framework for training models to differentiate between samples, which is versatile, and suitable for both supervised learning Khosla et al. (2020) and self-supervised learning (SSL). Contrastive SSL methods utilize the InfoNCE-based loss to draw samples (positive keys) closer to a query sample if they are similar, and to push away samples (negative keys) that are dissimilar, all within a designated embedding space. These methods often require large batch sizes for an effective contrasting Chen et al. (2020), additional memory banks for momentum updates He et al. (2020), or additional negative sampling strategies Hu et al. (2021). In our study, we use an InfoNCE-based contrastive loss, applied to the output logit space, to address SFDA challenges.

## 3 Source-Free Domain Adaptation

In this section, we outline the problem setup for SFDA and explain the implementation of contrastive learning to solve it. We also define the concept of *neighborhood* in our study, a key element for our theoretical analysis and proposed method.

### 3.1 Problem Statement

In customized applications, users typically possess their own data and a model pre-trained on a large dataset such as ImageNet Russakovsky et al. (2015), but do not have access to the model's original training data. SFDA methods aim to address this by enabling the model to adapt to the new data domain (*target domain*) without needing access to the original training data domain (*source domain*).

**Pre-training on a source domain**  We define a dataset from a source domain as $\mathcal{D}_S := \{(\mathbf{x_s}^i, \mathbf{y_s}^i)\}_{i=1}^N$ where $\mathbf{x_s}^i$ denotes the $i$-th data sample from the source dataset, and $\mathbf{y_s}^i$ is its corresponding class label. The objective of the pre-training stage is to derive a model $f_s := F_s \circ G_s$ (referred to as the *source model*) that minimizes the classification error on the source data $\epsilon_{\mathcal{D}_S}(f_s) := \sum_{i=1}^N \mathbb{P}[f_s(\mathbf{x}_s^i) \neq \mathbf{y}_s^i]$, where $G$ is a feature extractor that encodes input samples into latent features, and $F$ is a task-specific classifier.

**Adaptation on a target domain**  Consider a dataset sampled from a target domain, denoted as $\mathcal{D}_T := \{\mathbf{x_t}^i\}_{i=1}^M$, where $\mathbf{x_t}^i$ refers to the $i$-th data sample from the target dataset. During adaptation, it is important to note that class labels for all target samples are not accessible. In this stage, the goal is to have a target model $f_t$ that minimizes the generalization error on the target data $\epsilon_{\mathcal{D}_T}(f_t) := \sum_{i=1}^M \mathbb{P}[f_s(\mathbf{x}_t^i) \neq \mathbf{y}_t^i]$, possibly guided by the source pre-trained model $f_s$ without having access to the source data. One example of leveraging the source pre-trained model $f_s$ is to initialize the parameters of the target model $f_t$ with those of $f_s$.

### 3.2 Contrastive SFDA

Contrastive learning is a widely recognized algorithm for SSL. It operates on the principle that if data samples can be effectively clustered within a specific embedding space, the resulting model has the potential to exhibit robust performance across various downstream tasks, provided that fine-tuning is conducted appropriately.

In the context of SFDA, we choose to formulate the contrastive objective within the output logit space instead of the feature or embedding space used in SSL. This decision is motivated by the use of equal number of classes for classification in both the source and target domains. Note that in our empirical findings, there is no noticeable difference between performing distribution alignment on the output logit space and the output probability space. To simplify the mathematical notations, we conduct all theoretical analyses focusing on the alignment of output logits. For clarity, the output probability vector is the softmax output of logit vector. For any input $\mathbf{x}$, the sum of all elements in the output probability vector equals one, i.e., $\sum_{z=1}^Z [Softmax(f_t(\mathbf{x}))]_z = 1$, where $Z$ denotes the number of classes.

In comparison to SSL, formulating in either the logit or probability space simplifies the direct clustering of data samples according to the number of ground truth classes.

### 3.3 Definition of Neighborhood

To establish a neighborhood, we first define a feature bank $\mathbb{B}_T := \{G(\mathbf{x_t}^i) \,|\, \mathbf{x_t}^i \in \mathcal{D}_T\}_{i=1}^M$ where $G$ is a feature extractor initialized with the parameters of $G_s$ and updated throughout the adaptation. Similar to the neighborhood discovery used in unsupervised representation learning Huang et al. (2019); Van Gansbeke et al. (2020), given a query sample, we define a neighborhood of the sample as its $K$-Nearest Neighbors ($K$-NNs), with proximity determined by the cosine similarity between the features of the query sample and those of another sample drawn from $\mathbb{B}_T$:

$$d(\mathbf{x}_t^i, \mathbf{x}_t^j) := \frac{G(\mathbf{x}_t^i) \cdot \mathbb{B}_{T,j}}{||G(\mathbf{x}_t^i)|| \cdot ||\mathbb{B}_{T,j}||}. \tag{1}$$

where $\mathbb{B}_{T,j}$ denotes the $j$-th element of $\mathbb{B}_T$ corresponding to the features of a data sample $\mathbf{x}_t^j$. Thus, the neighborhood $\mathbb{N}_K(\mathbf{x}_t^i)$ of $\mathbf{x}_t^i$ comprises the top $K$ similar samples with respect to feature similarity, defined as:

$$\mathbb{N}_K(\mathbf{x}_t^i) := \arg max_{\mathcal{S} \subset \mathbb{B}_T, |\mathcal{S}|=K} \sum_{\mathbf{x} \in \mathcal{S}} d(\mathbf{x}_t^i, \mathbf{x}). \tag{2}$$

The centroid of the neighborhood, denoted as $\mu_K(\mathbf{x}^i)$, is the mean of the feature vectors of each sample within $\mathbb{N}_K(\mathbf{x}_t^i)$.

# 4 Theoretical Analysis of Contrastive SFDA

In this section, we engage in theoretical analysis concerning alignment errors when performing contrastive learning in the output logit space. In the meantime, we forge links between these error terms and the ground truth labels of target samples in the context of SFDA. This theoretical analysis not only sheds light on the crucial aspects of SFDA that necessitate attention from the research community, but it also clarifies the existing research gaps that our work endeavors to fill. All the proofs can be found in the appendix.

## 4.1 The Behavior of Contrastive Loss in SFDA

We start by closely examining how InfoNCE-based contrastive loss can be applied to address SFDA problems. For brevity, we use the term *transformation* to encompass the process of generating positive samples $\mathbf{x}^+$ from the query $\mathbf{x}$, whether through neighborhood searching or data augmentation. The transformations of other samples within the same mini-batch are used as negative samples $\mathbf{x}^-$.

Cosine similarity is commonly used in SSL to formulate contrastive loss within a high-dimensional embedding space. However, when applying contrastive loss in the output logit space for SFDA, further clarification is necessary. To provide a clearer understanding of this modification, we begin our discussion by redefining an upper bound for the contrastive loss formulated in the output logit space. This refinement allows us to focus on analyzing the alignment of output predictions through contrastive loss, using Euclidean distance as our measure.

**Proposition 1.** *When formulated in the output logit space, the InfoNCE-based contrastive loss, denoted as $\mathcal{L}_{cont}$, serves as an upper bound for the misalignment associated with two distinct alignment errors in predictions.*

$$\sum_{i=1}^{m}\left(\log(m-1)+\frac{\|f_t(\mathbf{x}_i)-f_t(\mathbf{x}_i^+)\|_2^2}{2\tau}-\frac{1}{m-1}\sum_{j\neq i}\frac{\|f_t(\mathbf{x}_i)-f_t(\mathbf{x}_j^+)\|_2^2}{2\tau}\right)\leq\mathcal{L}_{cont}.$$

Here, $\tau$ represents the temperature parameter, $m$ is the mini-batch size, $\mathbf{x}_j^+$ denotes a negative sample $\mathbf{x}^-$ given the query $\mathbf{x}_i$, and $\mathcal{L}_{cont}$ is defined as follows:

$$\mathcal{L}_{cont}=-\sum_{i=1}^{m}\log\frac{e^{f_t^\top(\mathbf{x}_i)f_t(\mathbf{x}_i^+)/\tau}}{\sum_{j\neq i}e^{f_t^\top(\mathbf{x}_i)f_t(\mathbf{x}_j^+)/\tau}}.$$

The proof for the proposition is provided in Appendix B.1. According to Proposition 1, minimizing the InfoNCE loss involves reducing the alignment error between the predictions of the query sample and its positive key $\mathbf{x}^+$, while introducing misalignment with the predictions of transformations applied to other samples within the same mini-batch. Therefore, optimizing the InfoNCE loss formulated in the logit space leads to the formation of discriminative clusters, however, it does not guarantee the alignment of these clusters with the ground truth.

## 4.2 What Has Been Missing in Contrastive SFDA

Now, let us begin establishing a connection between the prediction alignments introduced by contrastive loss and the target classification error in a task involving $Z$ classes. With $z$ denoting an index, which corresponds to a specific ground truth class, for a group of output logits. $\mathcal{C}_z$ represents a set of target samples belonging to class $z$ and having predictions that are close to each other in the output logit space. Within each $\mathcal{C}_z$, there exists a subset $\mathcal{C}_z^\delta$ that exclusively comprises samples predicted to be class $z$ and exhibits no overlapping with any other $C_l$ for $l\neq z$. Therefore, we can reasonably assume that there exists an Euclidean space in

which the logits of this subset of samples can be enclosed within a ball $\mathcal{C}_z^\delta$ of diameter $\delta$. The set of positive keys, whose logits can confidently be considered to lie within this ball, can be represented as:

$$\mathcal{S}_z^+ = \{\mathbf{x}^+ \in \mathcal{C}_z : \forall \mathbf{x} \in \mathcal{C}_z^\delta, \|f_t(\mathbf{x}^+) - f_t(\mathbf{x})\|_2^2 \leq \delta\},$$

and a set of negative keys, whose logits are confirmed to exist outside the ball, is denoted as:

$$\mathcal{S}_z^- = \{\mathbf{x}^- \in \mathcal{C}_z : \forall \mathbf{x} \in \mathcal{C}_z^\delta, \|f_t(\mathbf{x}^-) - f_t(\mathbf{x})\|_2^2 > \delta\}.$$

To establish a coherent connection between the target classification error and the prediction alignment errors, we introduce a definition for the target classification error concerning the logit groups and their associated ground truth:

$$\epsilon_{\mathcal{D}_T} = \sum_{z=1}^{Z} (\mathbb{P}[f_t(\mathbf{x}_t) \neq z, \forall \mathbf{x}_t \in \mathcal{C}_z] + \mathbb{P}[z \neq \mathbf{y}_t, \forall \mathbf{x}_t \in \mathcal{C}_z]), \tag{3}$$

The first term represents the probability of misclassification by the classifier $f_t$ for a given group class $z$ within the logit group $\mathcal{C}_z$. The second term indicates the probability that the ground truth for a sample $\mathbf{x}_t$ within the logit group $\mathcal{C}_z$ does not align with the group class $z$. This notation for classification error is inspired by the theoretical framework developed within the context of contrastive SSL Huang et al. (2021b).

Assuming that near-perfect prediction alignments, as introduced in Proposition 1, can be achieved by minimizing $\mathcal{L}_{cont}$, we can derive an upper bound for the error defined in Equation 3 as follows:

**Lemma 2.** *If $\mathcal{C}_z^\delta \cap \mathcal{C}_l^\delta = \emptyset$ holds for any $l \neq z$, then the error $\epsilon_{\mathcal{D}_T}$ defined on the groups of logits is upper bounded by:*

$$\epsilon_{\mathcal{D}_T} \leq R_\delta + \sum_{z=1}^{Z} \left( \mathbb{P}[f_t(\mathbf{x}^+) \neq \mathbf{y}_t, \forall \mathbf{x}^+ \in \mathcal{S}_z^+] + \mathbb{P}[f_t(\mathbf{x}^-) = \mathbf{y}_t, \forall \mathbf{x}^- \in \mathcal{S}_z^-] \right),$$

*where $R_\delta = \frac{\cup_{z=1}^{Z}(\mathcal{C}_z - \mathcal{C}_z^\delta)}{\cup_{z=1}^{Z} \mathcal{C}_z}$ and $(\mathcal{C}_z - \mathcal{C}_z^\delta)$ may overlap with $(\mathcal{C}_l - \mathcal{C}_l^\delta)$ for any $l \neq z$.*

The proof for the lemma is provided in Appendix B.2. Lemma 2 suggests that the target classification error, as defined in Equation 3, is affected by the overlap between $\mathcal{C}_z$ and $\mathcal{C}_l$ for any $l \neq z$, as well as the misclassification of the transformations $\mathbf{x}^+$ and $\mathbf{x}^-$. Given the connection between the prediction alignments introduced by contrastive loss and the target classification error, we will now explore how the two types of transformations, namely, *neighborhood searching* and *augmentation*, contribute to reducing the upper bound provided in Lemma 2.

**Data augmentation.** Augmentation-based contrastive SFDA methods augment data in the input space and use the logits of the augmented view as the positive key for contrastive clustering Zhang et al. (2022). However, solely relying on the augmented view transformed in the input space has limitations in mitigating the overlap of clusters Huang et al. (2021b). Standard data augmentation without distribution alignment in a contrastive SFDA framework may not effectively address misclassifications of positive keys due to insufficient information about the target ground truth – the **first oversight**.

**Neighborhood searching.** It involves using the logits of the query's $K$ nearest neighbors, located in the latent feature space *w.r.t* the model parameters, as the positive key for contrastive clustering. However, mitigating the misclassification of transformations through neighborhood searching, akin to $K$-NN classifiers, requires an intensive search to find the most suitable $K$ for the given task and data. Increasing the number of nearest neighbors, *i.e.,* larger $K$, results in smoother predictions but also leads to a greater overlap of logit groups, *i.e.,* higher $R_\delta$ Wu et al. (2002) – the **second oversight**. Existing methods Yang et al. (2022) assume that the informativeness of a query's neighborhood can be achieved by initializing the model $f$ with the parameters of the source model $f_s$. However, as the model is updated for contrastive clustering, the initially informative neighborhood information from $f_s$ diminishes. This underscores the necessity for a more effective utilization of $f_s$ during adaptation to enhance the informativeness of query neighborhoods and to further reduce the risk of misclassifying transformations – the **third oversight**.

In summary, existing contrastive SFDA methods can mitigate either $R_\delta$ (with data augmentation) or the misclassification of transformations (with neighborhood searching), but they cannot handle both of them.

# 5 The Proposed Framework

In this section, we present our *source-informed latent augmented neighborhood* (SiLAN), which leverages the advantages of both *neighborhood searching* and *augmentation* to address the previously mentioned oversights.

## 5.1 Source-informed Latent Augmentation

To clarify the proposed work, we introduce an additional feature bank denoted as $\mathbb{B}_S := \{G_s(\mathbf{x_t}^i) \,|\, \mathbf{x_t}^i \in \mathcal{D}_T\}_{i=1}^M$. During adaptation, $G_s$ remains unchanged and is used to search the source-informed neighborhood $\mathbb{N}_K^s(\mathbf{x_t}^i)$ for a given target query $\mathbf{x_t}^i$. For a given $\mathbf{x_t}^i$, we independently and identically (i.i.d) sample random noise $\xi \in \mathbb{R}^H$ from $\mathcal{N}(\mathbf{0}, \sigma_K^{s\,2}(\mathbf{x_t}^i))$, where $\sigma_K^{s\,2}(\mathbf{x_t}^i)$ is the variance of $\mathbb{N}_K^s(\mathbf{x_t}^i)$, and $H$ is the dimension of a feature vector. We then add $\xi$ to the latent features of the query's neighborhood (*w.r.t* the current target model $f_t$) centroid, denoted as $\mu_K^t(\mathbf{x_t}^i)$, to generate positive keys:

$$\hat{\mathbf{h}} := G_t(\mu_K^t(\mathbf{x_t}^i)) + \xi. \tag{4}$$

Note that $\hat{\mathbf{h}}$ follows a Gaussian distribution, *i.e.*, $\hat{\mathbf{h}} \sim \mathcal{N}(G_t(\mu_K^t(\mathbf{x_t}^i)), \sigma_K^{s\,2}(\mathbf{x_t}^i))$.

## 5.2 InfoNCE-based Contrastive Framework

To illustrate the integration of SiLAN into existing contrastive SFDA frameworks, we specifically employ the InfoNCE contrastive framework Oord et al. (2018); Chen et al. (2020); He et al. (2020) as an example. We adhere to the settings of existing contrastive SFDA methods, formulating it within the output logit space for each mini-batch of size $m$:

$$\mathcal{L}_{SiLAN} = -\sum_{i=1}^m log \frac{e^{[F_t(G_t(\mathbf{x_t}^i))]^\top F_t(\hat{\mathbf{h}}_i)/\tau}}{\sum_{j \neq i} e^{[F_t(G_t(\mathbf{x_t}^i))]^\top F_t(\hat{\mathbf{h}}_j)/\tau}} \tag{5}$$

where $G_t$ and $F_t$ represent the feature extractor and task-specific classifier of $f_t$, respectively. $\hat{\mathbf{h}}_j$ denotes the augmentation of other samples in the same mini-batch, and $\tau$ is the temperature parameter. Note that the focus of our work is to examine the informativeness of the dispersion of $\mathbb{N}_K^s$ in the latent space. Consequently, we refrain from incorporating additional techniques, such as momentum update He et al. (2020), to enhance contrastive clustering, despite their potential to improve the performance.

From an intuitive perspective, optimizing a contrastive objective with positive keys defined by $F_t(\hat{\mathbf{h}}_i)$ can be likened to applying forces that attract the predictions of $\hat{\mathbf{h}}_i$, which are informed by $\mathbb{N}_K^s(\mathbf{x_t}^i)$, towards the logit group where the query sample $\mathbf{x_t}^i$ belongs to. In the meantime, this optimization exerts a repelling effect on those $\hat{\mathbf{h}}_j$ that are located outside the logit group, pushing them away from it.

## 5.3 Integrating SiLAN into InfoNCE during Adaptation

To clarify the integration of our SiLAN into the InfoNCE contrastive framework, we provide a detailed description of the integration process in Algorithm 1, which outlines the target adaptation phase when incorporating SiLAN.

## 5.4 Unveiling Rationale for Latent Augmented Neighborhood

Diversifying data augmentation techniques in the input space enhances the concentration of augmented data Wang & Liu (2021), thereby reducing $R_\delta$. Yet, it introduces notable variations in the latent space due to minor changes in input, posing a risk of altering the semantics of queries. To address this, we propose performing latent augmentation on the query's neighborhood centroid for positive key generation.

To understand the rationale behind our latent augmentation on the query's neighborhood centroid, let us first examine a general case where random noise $\xi_0 \sim \mathcal{N}(\mathbf{0}, \sigma^2)$ is added to the latent features of $\mu_K(\mathbf{x})$. Such augmentation enriches the diversity of the neighborhood by allowing the transformation to traverse around a Gaussian profile whose radius is determined by its standard deviation $\sigma$. A larger $\sigma$ promotes extensive exploration, capturing diverse but potentially irrelevant data variations. Conversely, a smaller $\sigma$

---

**Algorithm 1:** SiLAN-Enhanced InfoNCE for Target Adaptation

---

**Input:** mini-batch target data $\mathbf{x_t} \in \mathcal{D}_T$; source model $F_s(G_s(\cdot))$; neighbor number $K$; temperature $\tau$ and maximum epoch $N$.

1   **Initialization:** $F_t(G_t(\cdot)) \leftarrow F_s(G_s(\cdot))$ and $n = 0$ ;

2   **while** $n \leq N - 1$ **do**

3      Find $K$-NNs for $\mathbf{x_t}$ using $G_t$ to form their target neighborhoods $\mathbb{N}_K^t(\mathbf{x_t})$;

4      Find $K$-NNs for $\mathbf{x_t}$ using $G_s$ to form their source-informed neighborhoods $\mathbb{N}_K^s(\mathbf{x_t})$;

5      Compute centroids for target neighborhoods: $\mu_K^t(\mathbf{x_t}) = \frac{1}{K}\sum_{i=1}^K \mathbb{N}_K^{t,i}(\mathbf{x_t})$;

6      Compute variances for source-informed neighborhoods:

$$\sigma_K^{s\,2}(\mathbf{x_t}) = \frac{1}{K}\sum_{j=1}^K \left( G_s\big(\mathbb{N}_K^{s,j}(\mathbf{x_t})\big) - \frac{1}{K}\sum_{i=1}^K G_s\big(\mathbb{N}_K^{s,i}(\mathbf{x_t})\big)\right)^2;$$

7      Sample noises from a Gaussian with variance $\sigma_K^{s\,2}(\mathbf{x_t})$: $\xi \sim \mathcal{N}(0, \sigma_K^{s\,2}(\mathbf{x_t}))$;

8      Augment the latent features for the positive keys to guide contrastive clustering:

$$\hat{\mathbf{h}} = G_t(\mu_K^t(\mathbf{x_t})) + \xi;$$

9      Optimize the model parameters for $F_t(G_t(\cdot))$ to minimize the InfoNCE loss $\mathcal{L}_{SiLAN}$:

$$-\sum_{i=1}^m log \frac{e^{[F_t(G_t(\mathbf{x_t^i}))]^\top F_t(\hat{\mathbf{h}}_i)/\tau}}{\sum_{j\neq i} e^{[F_t(G_t(\mathbf{x_t^i})]^\top F_t(\hat{\mathbf{h}}_j)/\tau}};$$

10      $n = n + 1$;

11   **end**

---

restricts exploration to local and fine-grained variations within a specific logit group. This flexibility enables the model to adapt to varying degrees of data complexity and distribution shifts, making it well-suited for reducing misclassification of the transformation in contrastive SFDA.

Moreover, traversing augmentations around Gaussian profiles enhances contrastive clustering by pushing non-overlapping logit group regions apart, ensuring well-separated features. This improved separability aligns with the desired behavior as established by the theoretical findings in the subsequent lemma. To be specific, for a query sample within a logit group $\mathcal{C}_i$, traversing its augmentations around the Gaussian profile (while aligning their predictions with that of the query through a contrastive objective) will stretch the non-overlapping region $\mathcal{C}_i^\delta$ and extend its boundary. Conversely, traversing the augmentations of its negative samples, which likely do not belong to the same category (diverging their predictions from that of the query), will push $\mathcal{C}_i^\delta$ farther away from other logit groups to which these negative samples belong, as illustrated in Figure 2. This adaptive process encourages the model $f_t$, optimized by a contrastive loss, to dynamically reduce $R_\delta$ during adaptation.

To clarify the subsequent lemma, consider $G_t^{opt}$ as a target feature extractor that converges on a contrastive objective with augmentation traversing a Gaussian profile with $\sigma$. To quantify the gap between clusters post-convergence, we employ the concept of calculating the effective region of a Gaussian beam as introduced in Hogg & Lang (2013). Following their approach, we denote the noise variance per augmentation in the extraneous noise as $\sigma_{ext}^2$ [1] from the overlapping regions.

**Proposition 3.** *If $\mathcal{C}_z^\delta \cap \mathcal{C}_l^\delta = \emptyset$ holds for any $z \neq l$, and the assumption that the representation of a query sample in the feature space remains close to those of its positive augmentations and far away from those of its negative samples holds, then for all $z \neq l$ and $i \neq j$ (where $\mathbf{x}_i \in \mathcal{C}_z^\delta$ and $\mathbf{x}_j \in \mathcal{C}_l^\delta$), their distance in the latent feature space has a lower bound as we take the limit $\sigma_{ext}^2 \to 0$, as follows:*

$$|G_t^{opt}(\mathbf{x}_i) - G_t^{opt}(\mathbf{x}_j)| \geq 3.1704\sigma.$$

The proof for the proposition is provided in Appendix B.3. Here, the optimal aperture radius for a Gaussian profile approximates $1.5852\sigma$. In the optimal scenario, the Gaussian profile maximizes its transformation-to-noise ratio under the assumption that $\sigma_{ext}^2$ is close to 0.

---

[1]Extraneous noise is a term derived from physics that refers to a transient portion of a wave, which does not belong to the ambient waves, nor does it originate from the source being studied. It is important to distinguish this concept from the notion of random noise used for augmentation.

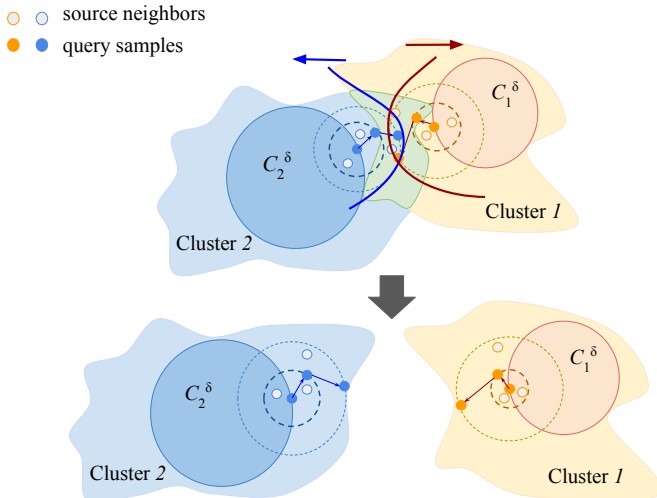

Figure 2: The transformation of a query sample traverses a Gaussian profile defined by source-informed neighbors in the latent space, initiating a pull-and-push effect to reduce feature overlaps among logit groups.

Similarly, let $F_t^{opt}$ denote the target classification head, which is linear in our work. With Proposition 3 in consideration, we can derive a lower bound for the gap between any two logit groups in the output space when a Gaussian profile is used to augment the query's neighborhood features:

**Lemma 4.** $\forall z \neq l$ and $i \neq j$, if the linear classifier $F^{opt}$ is L-bi-Lipschitz continuous, $\mathcal{C}_z^\delta \cap \mathcal{C}_l^\delta = \emptyset$ holds for any $z \neq l$ where $\mathbf{x}_i \in \mathcal{C}_z^\delta$ and $\mathbf{x}_j \in \mathcal{C}_l^\delta$, and $\sigma_{ext}^2 \to 0$:

$$|f_t^{opt}(\mathbf{x}_i) - f_t^{opt}(\mathbf{x}_j)| \geq \frac{3.1704\sigma}{L},$$

where $f_t^{opt} = F_t^{opt}(G_t^{opt}(\cdot))$ and the Lipschitz constant $L$ depends on the number of parameters of $F_t(\cdot)$.

The proof for the lemma is provided in Appendix B.4. Lemma 4 suggests that when the parameters of $f_t(\cdot)$ converge on the contrastive loss with SiLAN augmentation for positive keys, the augmented views of the query's neighborhood centroid within a Gaussian profile can ensure a lower bound, dependent on the standard deviation of the Gaussian and the number of parameters of the linear classifier, for the minimum gap among the non-overlapping regions of the logit groups.

Therefore, we can conclude that a better separation among logit groups (larger non-overlapping regions) can be achieved by using a Gaussian profile with a higher standard deviation for SiLAN, as stated in Lemma 4. However, if the Gaussian profile extends beyond the boundaries of the query's logit group in the latent feature space (when $\sigma$ is too large), the augmentation might traverse outside the logit group. This introduces ambiguity into contrastive clustering, leading to more errors in mislabeling transformations. To be specific, a large $\sigma$ for the augmentation might cause the predictions of positive keys to fall outside the query's logit group, leading to an increase in $\sum_{z=1}^{Z}(\mathbb{P}[f_t(\mathbf{x}^+) \neq \mathbf{y}_t, \forall \mathbf{x}^+ \in \mathcal{S}_z^+])$. Meanwhile, it could result in the predictions of negative keys falling within the query's logit group, leading to an increase in $\mathbb{P}[f_t(\mathbf{x}^-) = \mathbf{y}_t, \forall \mathbf{x}^- \in \mathcal{S}_z^-])$.

## 5.5 Informativeness of $\sigma$

Recalling the phenomenon highlighted in *Observations on Neighborhood Informativeness*, which suggests that the dispersed neighbors derived from $\mathbb{N}_k^s(\mathbf{x}_t)$ carry valuable insights into the ground truth of $\mathbf{x}_t$ (as nearby points share the same labels), the incorporation of this information becomes significant in the process of discriminative clustering. Therefore, it is crucial to perform SILAN in a manner that covers these neighbors to reduce the likelihood of mislabeled augmentations. To achieve this, expanding the scope of the Gaussian profile's influence based on the query's neighborhood as determined by the source model is

necessary. Specifically, the radius of the Gaussian profile should be determined by $\mathbb{N}_K^s(\mathbf{x_t})$, thereby setting $\sigma$ to $\sigma_K^s$.

In summary, the optimal value for $\sigma$ should be large enough to create a substantial gap among logit groups, making them more distinguishable, while avoiding the generation of an excessive number of ambiguous augmentations. Interestingly, the dispersion of $\mathbb{N}_K^s(\mathbf{x}_t)$ that adversely impacts discriminability on $\mathcal{D}_T$ actually proves beneficial in determining the optimal value of $\sigma$ for the proposed SiLAN.

# 6 Experiments

## 6.1 Experiments on Toy Dataset

In this section, we demonstrate the effectiveness of SiLAN on a toy dataset, the *moon* dataset, which simulates domain shift by rotating data sample orientations. The source domain features an interleaving half circle with 1000 data points, and the target domain replicates this structure with a 30-degree rotation around the mean. Both domains incorporate Gaussian noise with a standard deviation of 0.1 and are generated using distinct random seeds. The experimental setup includes a 5-layer fully connected network. As depicted in Figure 3, the source-only model encounters challenges in accurately classifying numerous points, while our SiLAN method excels in achieving accurate classification.

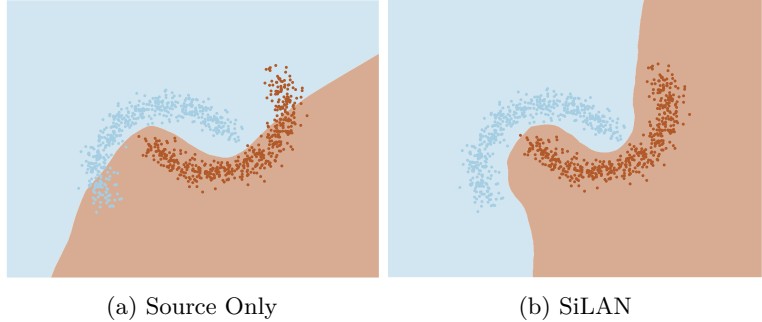

(a) Source Only          (b) SiLAN

Figure 3: **(Best viewed in color.)** Decision boundaries derived from *moon* dataset *w/wo* SiLAN.

Table 1: Comparison of SFDA methods using ResNet-50 on *Office-31*. The best results are highlighted.

| Method | A→D | A→W | D→W | D→A | W→D | W→A | Avg. |
|---|---|---|---|---|---|---|---|
| ResNet-50 He et al. (2016) | 68.9 | 68.4 | 96.7 | 62.5 | 99.3 | 60.7 | 76.1 |
| SHOT Liang et al. (2020) | 94.0 | 90.1 | 98.4 | 74.7 | 99.9 | 74.3 | 88.6 |
| NRC Yang et al. (2021a) | 96.0 | 90.8 | 99.0 | 75.3 | 100.0 | 75.0 | 89.4 |
| 3C-GAN Li et al. (2020) | 92.7 | 93.7 | 98.5 | 75.3 | 99.8 | **77.8** | 89.6 |
| HCL Huang et al. (2021a) | 94.7 | 92.5 | 98.2 | 75.9 | 100.0 | 77.7 | 89.8 |
| AaD Yang et al. (2022) | 96.4 | 92.1 | **99.1** | 75.0 | 100.0 | 76.5 | 89.9 |
| SF(DA)$^2$ Hwang et al. (2024) | 95.8 | 92.1 | 99.0 | 75.7 | 99.8 | 76.8 | 89.9 |
| **SiLAN (Ours)** | **97.1** | **95.8** | 98.9 | **76.4** | **100.0** | 76.9 | **90.7** |

Table 2: Comparison of the SFDA methods on *Office-Home* (ResNet-50).

| Method | Ar→Cl | Ar→Pr | Ar→Rw | Cl→Ar | Cl→Pr | Cl→Rw | Pr→Ar | Pr→Cl | Pr→Rw | Rw→Ar | Rw→Cl | Rw→Pr | Avg. |
|---|---|---|---|---|---|---|---|---|---|---|---|---|---|
| ResNet-50 He et al. (2016) | 34.9 | 50.0 | 58.0 | 37.4 | 41.9 | 46.2 | 38.5 | 31.2 | 60.4 | 53.9 | 41.2 | 59.9 | 46.1 |
| G-SFDA Yang et al. (2021b) | 57.9 | 78.6 | 81.0 | 66.7 | 77.2 | 77.2 | 65.6 | 56.0 | 82.2 | 72.0 | 57.8 | 83.4 | 71.3 |
| SHOT Liang et al. (2020) | 57.1 | 78.1 | 81.5 | 68.0 | 78.2 | 78.1 | 67.4 | 54.9 | 82.2 | 73.3 | 58.8 | 84.3 | 71.8 |
| NRC Yang et al. (2021a) | 57.7 | 80.3 | 82.0 | 68.1 | **79.8** | 78.6 | 65.3 | 56.4 | **83.0** | 71.0 | 58.6 | 85.6 | 72.2 |
| AaD Yang et al. (2022) | 59.3 | 79.3 | 82.1 | 68.9 | **79.8** | 79.5 | 67.2 | 57.4 | 83.1 | 72.1 | 58.5 | 85.4 | 72.7 |
| DaC Zhang et al. (2022) | **59.5** | 79.5 | 81.2 | 69.3 | 78.9 | 79.2 | 67.4 | 56.4 | 82.4 | 74.0 | **61.4** | 84.4 | 72.8 |
| SF(DA)$^2$ Hwang et al. (2024) | 57.8 | 80.2 | 81.5 | 69.5 | 79.2 | 79.4 | 66.5 | 57.2 | 82.1 | 73.3 | 60.2 | 83.8 | 72.6 |
| **SiLAN (Ours)** | 58.2 | **81.2** | **82.5** | **69.8** | 78.6 | **80.3** | **68.4** | **58.6** | 82.5 | **75.6** | 60.8 | **86.1** | **73.6** |

Table 3: Comparison of the SFDA methods on *VisDA2017* (ResNet-101).

| Method | plane | bcycl | bus | car | horse | knife | mcycl | person | plant | sktbrd | train | truck | Avg. |
|---|---|---|---|---|---|---|---|---|---|---|---|---|---|
| ResNet-101 He et al. (2016) | 55.1 | 53.3 | 61.9 | 59.1 | 80.6 | 17.9 | 79.7 | 31.2 | 81.0 | 26.5 | 73.5 | 8.5 | 52.4 |
| SHOT Liang et al. (2020) | 94.3 | 88.5 | 80.1 | 57.3 | 93.1 | 94.9 | 80.7 | 80.3 | 91.5 | 89.1 | 86.3 | 58.2 | 82.9 |
| HCL Huang et al. (2021a) | 93.3 | 85.4 | 80.7 | 68.5 | 91.0 | 88.1 | 86.0 | 78.6 | 86.6 | 88.8 | 80.0 | 74.7 | 83.5 |
| G-SFDA Yang et al. (2021b) | 96.1 | 88.3 | 85.5 | 74.1 | 97.1 | 95.4 | 89.5 | 79.4 | 95.4 | 92.9 | 89.1 | 42.6 | 85.4 |
| NRC Yang et al. (2021a) | 96.8 | **91.3** | 82.4 | 62.4 | 96.2 | 95.9 | 86.1 | 80.6 | 94.8 | 94.1 | 90.4 | 59.7 | 85.9 |
| AaD Yang et al. (2022) | 95.2 | 90.5 | 85.5 | 79.2 | 96.4 | 96.2 | 88.8 | 80.4 | 93.9 | 91.8 | 91.1 | 55.9 | 87.1 |
| DaC Zhang et al. (2022) | 96.6 | 86.8 | **86.4** | 78.4 | 96.4 | 96.2 | **93.6** | **83.8** | **96.8** | 95.1 | 89.6 | 50.0 | 87.3 |
| SF(DA)$^2$ Hwang et al. (2024) | 96.8 | 89.3 | 82.9 | 81.4 | 96.8 | **95.7** | 90.4 | 81.3 | 95.5 | 93.7 | 88.5 | **64.7** | 88.1 |
| **SiLAN (Ours)** | **97.5** | 90.1 | 85.8 | **80.4** | **97.6** | 95.5 | 92.0 | 82.9 | 96.5 | **95.3** | **92.6** | 53.4 | **88.3** |

## 6.2 Experiments on Benchmark Datasets

**Datasets.** We have evaluated our SiLAN on three benchmark datasets for SFDA:

- **Office-31** Saenko et al. (2010), which consists of 4,652 images of 31 object classes captured from three domains: Amazon (**A**), Webcam (**W**), and DSLR (**D**).

- **Office-Home** Venkateswara et al. (2017), which has 15,500 images of 65 classes from four domains: Artistic (**Ar**), Clipart (**Cl**), Product (**Pr**), and Real-World (**Rw**).

- **VisDA-C 2017** Peng et al. (2017), a large-scale dataset used for the 2017 ICCV visual DA challenge, with 280K images of 12 object categories. The source domain contains synthetic images generated via 3D model rendering, while the target domain consists of real images.

**Experiment Setup.** For fair comparisons with other SFDA methods, we maintain consistency in network architecture, training techniques, and hyperparameters across all experiments. Specifically, we utilize ResNet-50 for Office-31 and Office-Home, and ResNet-101 for VisDA-C. The classifier $F$ consists of two linear layers. We adopt the InfoNCE-based loss from *SimCLR* Chen et al. (2020) for contrastive learning. The optimization uses an SGD optimizer with a momentum of 0.9, and the batch size is set to 32 for Office datasets and 64 for VisDA-C. The learning rates for all experiments are fixed at $1e^{-3}$. Results for Office-31 and Office-Home are reported after 100-epoch training, while VisDA results are presented after 15-epoch training. In Office-Home experiments, we apply regularization to the diagonal matrix of predictions in a mini-batch, achieved through singular value decomposition Cui et al. (2020). The SiLAN hyperparameters, including $K_t$ and $K_s$, represent the number of neighbors determined by the target and source models, respectively. They are set to 3 for most Office experiments and 15 for VisDA-C. The temperature parameter $\tau$ for *InfoNCE* loss is set to 0.11 for most experiments.

**Results.** Results for Office-31, Office-Home, and VisDA-C 2017 are shown in Tables 1, 2, and 3, respectively. The results for *ResNet-X* represent applying the source pre-trained model directly to target domain without adaptation. Our framework achieves state-of-the-art performance across all three benchmarks compared to other SFDA methods.

## 7 Conclusions

The present paper provided a thorough analysis of SFDA methods based on contrastive learning, revealing that their effectiveness depends on the degree of overlap among logit groups and the informativeness of augmentations for positive key generation. Building on these insights, we introduced a novel latent augmentation method to address the limitations of existing contrastive SFDA methods. This method utilized the dispersion of latent features around query samples, guided by the source pre-trained model, to reduce logit group overlaps and improve positive key generation. The proposed method, relying on a single InfoNCE loss, demonstrated superior performance compared to existing SFDA methods across various benchmark datasets. Our research contributed to the understanding of applying contrastive learning in the context of SFDA and provides a practical solution to enhance model generalization in the absence of source data.

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

# A  Ablation Studies

## A.1  Number of Nearest Neighbors for $\mathbb{N}_K^t$ and $\mathbb{N}_K^s$.

During target adaptation, we distinguish the number of nearest neighbors for $\mathbb{N}_K^t$ determined by the target model as $K_t$ and for $\mathbb{N}_K^s$ determined by the source model as $K_s$. As discussed earlier, selecting a small value for $K_t$ can enhance clustering but may introduce less smooth predictions, susceptible to noise from inconsistent neighbors. Conversely, choosing a large value of $K_t$ yields smoother predictions but increases the overlap among logit groups. The value of $K_s$ should be determined empirically to find a balance. It should ensure a sufficiently large standard deviation for traversal around $\mathbb{N}_K^s$ while avoiding excessive values that introduce augmentations whose predictions lie in other logit groups. We conduct this ablation study on Office-31 to adapt the model pre-trained on *Amazon* to *Webcam*. As shown in Table 4, our framework is robust to the choices of both $K_t$ and $K_s$ within a reasonable range.

## A.2  Standard Deviation of $\mathbb{N}_K^s$.

The impact of $K_s$ on the standard deviation of latent features within $\mathbb{N}_K^s$ is evaluated through experiments on the Office-31 dataset. In this ablation study, a model trained on *Amazon* is adapted to *Webcam*. Table 4 presents the average standard deviations of the features extracted from all samples in *Webcam* using the converged model, with varying values of $K_s$, under the column **Noise Std**.

Table 4: Ablation study for the number of nearest neighbors, $K_t$ and $K_s$, on the adaptation performance on Office-31 using ResNet-50.

| Office-31 Amazon→Webcam | | | | | | | |
|---|---|---|---|---|---|---|---|
| $K_t$ | $K_s$ | Result | Noise Std. | $K_t$ | $K_s$ | Result | Noise Std. |
| 2 | 2 | 92.7 | 0.049 | 3 | 2 | 93.6 | 0.078 |
| 2 | 3 | 93.2 | 0.068 | **3** | **3** | **94.6** | **0.105** |
| 2 | 4 | 93.0 | 0.059 | 3 | 4 | 94.0 | 0.085 |
| 2 | 5 | 92.8 | 0.052 | 3 | 5 | 93.2 | 0.071 |
| 4 | 2 | 92.8 | 0.058 | 5 | 2 | 92.2 | 0.032 |
| 4 | 3 | 93.4 | 0.062 | 5 | 3 | 93.2 | 0.068 |
| 4 | 4 | 93.2 | 0.056 | 5 | 4 | 93.4 | 0.070 |
| 4 | 5 | 93.9 | 0.078 | 5 | 5 | 93.2 | 0.072 |

### A.3 Temperature for Contrastive Loss.

The strength of penalties on negative keys in the contrastive loss is governed by the temperature parameter $\tau$. A small temperature increases the penalization of negative samples, pushing their latent features farther away from those of the query, as highlighted in Wang & Liu (2021). Conversely, a large temperature results in more compact latent features within each logit group, but reduces sensitivity to negative samples in clustering. To empirically evaluate the effect of $\tau$, experiments are conducted on Office-31 with $K_s$ and $K_t$ set to 3, respectively. The results, illustrated in Figure 4, demonstrate the impact of $\tau$ on the target classification performance.

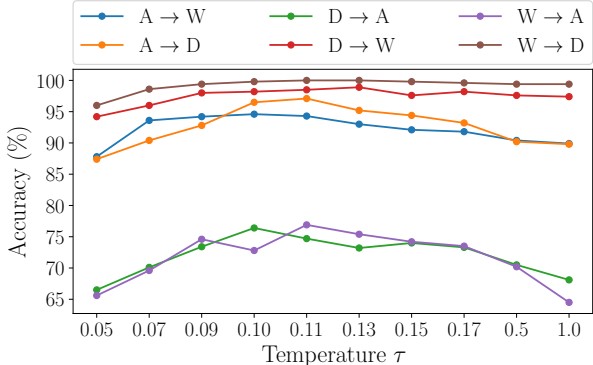

Figure 4: Ablation experiments on Office-31 using ResNet-50 to evaluate how classification performance varies with temperature parameter $\tau$.

### A.4 Ablation on SiLAN's Impact on InfoNCE Loss.

In this ablation study, we evaluate the effectiveness of SiLAN in improving the *InfoNCE* baseline using the Office-Home dataset. We compare the classification accuracies in the target domain for models trained with and without the use of SiLAN for positive key generation to guide *InfoNCE* contrastive clustering. The results, as shown in Table 5, demonstrate that SiLAN significantly improves the performance of the InfoNCE baseline, highlighting its robustness.

### A.5 General Guidance for Hyperparameter Selection

Our InfoNCE-based SiLAN introduces three additional hyperparameters: the number of the source-informed kNNs $K_s$, the number of target kNNs $K_t$, and the logit temperature for contrastive loss $\tau$. In this section, we

Table 5: Comparing the InfoNCE baseline performance with and without SiLAN on Office-Home using ResNet-50.

| Method | Ar → | | | Cl → | | |
|---|---|---|---|---|---|---|
| | Cl | Pr | Rw | Ar | Pr | Rw |
| InfoNCE+$K$-NNs | 55.6 | 76.4 | 80.6 | 66.4 | 75.2 | 76.4 |
| InfoNCE+SiLAN (Ours) | **58.2** | **81.2** | **82.5** | **69.8** | **78.6** | **80.3** |
| Method | Pr → | | | Rw → | | |
| | Ar | Cl | Rw | Ar | Cl | Pr |
| InfoNCE+$K$-NNs | 66.2 | 53.8 | 80.5 | 72.8 | 56.8 | 83.5 |
| InfoNCE+SiLAN (Ours) | **68.4** | **58.6** | **82.5** | **75.6** | **60.8** | **86.1** |

will offer general guidance for setting values for these hyperparameters based on the results of the sensitivity analysis obtained from Appendices A.1, A.2, and A.3.

To identify the optimal set of hyperparameters, we recommend the following systematic approach: begin by determining the number of target $k$NNs, denoted as $K_t$. This parameter determines the mean of the Gaussian for our latent augmentation, directly influencing the effectiveness of contrastive clustering. In general, $K_t$ can be roughly estimated based on the number of samples per class in the target dataset; a smaller target dataset should use a smaller $K_t$. For instance, our experiments show that the optimal $K_t$ ranges from 3 to 5 for Office datasets, each containing about 200 images per class. However, for VisDA-2017, which comprises around 23K images per class, the optimal $K_t$ is 15. Therefore, to determine the optimal $K_t$, we suggest a search range from 2 to 8 for a target dataset with 100 to 1K samples per class, and a search range from 10 to 20 for a target dataset with more than 20K images per class.

Subsequently, the number of the source-informed $k$NNs, denoted as $K_s$, which determines the standard deviation of the Gaussian for our SiLAN, could be decided based on $K_t$. Our analysis in Section 5 indicates that $K_s$ should be large enough to contain the farthest source-informed neighbor that may share the same ground truth as the target query. However, it should not be excessively large to avoid including features with inconsistent ground truth in the positive key generation. We found that the standard deviation of these source-informed kNNs' features is a reliable indicator for selecting $K_s$ to define the optimal latent augmentation region. As detailed in Appendix A.2, a larger standard deviation of the latent feature vectors of the source-informed kNNs generally correlates with better performance on target classification tasks. Typically, the largest standard deviation occurs when $K_s$ is approximately equal to $K_t$. Therefore, a preliminary strategy could be setting $K_s$ equal to $K_t$, with the optimal $K_s$ lying within the range of $K_t \pm 2$.

Finally, the temperature $\tau$ for contrastive logits should be determined similarly to self-supervised learning frameworks, as regardless of the mathematical space in which clustering occurs, this parameter influences the degree of penalization applied to hard negative samples Wang & Liu (2021). Unlike unsupervised representation learning, which aims for a universal representation across tasks, our contrastive objective operates in the output logit space, favoring a small value for $\tau$ (similar to identifying the temperature in knowledge distillation frameworks Hinton et al. (2015)). In our experiments, the optimal value for $\tau$ falls between 0.07 and 0.15. Therefore, a search range between 0.05 and 0.2 is recommended for practitioners.

## A.6  Computational Analysis

In this section, we conduct a detailed runtime analysis on our SiLAN and compare it with other advanced SFDA methods. To be specific, we conducted a runtime analysis comparing the one-epoch training time and overall convergence runtime of AaD Yang et al. (2022) of our SiLAN with other SFDA methods. To demonstrate scalability with respect to dataset size and model complexity, we performed this computational analysis on both the small-scale dataset Office-31 using ResNet-50 and the large-scale dataset VisDA-2017 using ResNet-101. All experiments were conducted on a machine equipped with an Nvidia V100 GPU.

Table 6: Time Analysis of One-Epoch Target Adaptation on Office-31 AD (ResNet-50).

| Method | One-Epoch Time (sec) | One-Epoch Performance (%) |
|---|---|---|
| NRC Yang et al. (2021a) | 21.4 | 85.3 |
| DaC Zhang et al. (2022) | 40.5 | 80.6 |
| AaD Yang et al. (2022) | 19.8 | 84.4 |
| SiLAN (ours) | 20.6 | 84.9 |
| AaD+SiLAN (ours) | 20.2 (+**0.4**) | 85.6 (+**1.2**) |

Table 7: Convergence Time Analysis for Target Adaptation on Office-31 AD (ResNet-50).

| Method | Convergence Time (sec) | Best Performance (%) |
|---|---|---|
| NRC Yang et al. (2021a) | 856.7 | 96.0 |
| DaC Zhang et al. (2022) | 1417.5 | 94.2 |
| AaD Yang et al. (2022) | 714.9 | 96.4 |
| SiLAN (ours) | 618.9 | 97.1 |
| AaD+SiLAN (ours) | 584.5 (−**130.4**) | 97.5 (+**1.1**) |

Table 8: Time Analysis of One-Epoch Target Adaptation on VisDA-2017 Dataset (ResNet-101).

| Method | One-Epoch Time (sec) | One-Epoch Performance (%) |
|---|---|---|
| NRC Yang et al. (2021a) | 469.2 | 79.2 |
| DaC Zhang et al. (2022) | 632.8 | 82.4 |
| AaD Yang et al. (2022) | 453.6 | 82.6 |
| SiLAN (ours) | 465.9 | 84.5 |
| AaD+SiLAN (ours) | 462.3 (+**8.7**) | 86.8 (+**4.2**) |

Moreover, as our SiLAN can also serve as a general latent augmentation method to enhance other SFDA methods, we have included a runtime analysis for the scenario where our SiLAN latent augmentation is applied to improve the informativeness of the latent features of the neighbors for AaD (referred to as *AaD+SiLAN*). In this analysis, we present the performance gain and additional runtime compared to AaD alone, providing practitioners with the information needed to balance performance gains against runtime considerations.

Tables 6, 7, 8, 9 illustrate that our SiLAN, NRC, and AaD exhibit similar target adaptation times per epoch, whereas DaC requires substantially more time due to its adaptive process and self-training steps, such as pseudo label generation and re-training. This observation aligns with the fact that AaD, NRC, and our SiLAN are all rooted in neighborhood searching and involve aligning predictions between query predictions and those of the neighbors. Meanwhile, our SiLAN, along with AaD and NRC, typically converge within about 10 epochs for the VisDA-2017 dataset, whereas DaC requires around 20 epochs. Notably, on the small-scale Office-31 dataset, our SiLAN achieves faster convergence compared to other methods. We hypothesize that this is because SiLAN, serving as an augmentation method, significantly enhances model convergence, particularly in situations where data is scarce. The total convergence time for target adaptation is also provided for comparison.

Therefore, we conclude that despite our SiLAN having a similar one-epoch runtime compared to NRC and AaD, its use as latent augmentation leads to faster convergence compared to other SFDA methods, resulting in overall runtime benefits.

## A.7  SiLAN as A General Latent Augmentation Method

Table 9: Convergence Time Analysis for Target Adaptation on VisDA-2017 Dataset (ResNet-101).

| Method | Convergence Time (sec) | Best Performance (%) |
|---|---|---|
| NRC Yang et al. (2021a) | 4692.8 | 85.9 |
| DaC Zhang et al. (2022) | 12656.3 | 87.3 |
| AaD Yang et al. (2022) | 4536.2 | 87.1 |
| SiLAN (ours) | 3727.2 | 88.3 |
| AaD+SiLAN (ours) | 3236.1 ($-$**1300.1**) | 89.7 ($+$**2.6**) |

In this ablation study, we demonstrate the versatility of our proposed SiLAN as a general latent augmentation method for source-free domain adaptation (SFDA) frameworks. To validate this, we incorporate SiLAN into advanced SFDA methods, namely HCL Huang et al. (2021a), A$^2$Net Xia et al. (2021), AaD Yang et al. (2022) and NRC Yang et al. (2021a). While both AaD and NRC utilize the predictions of query neighbors to optimize the model, their implementation details vary.

Among them, HCL Huang et al. (2021a) is a contrastive SFDA framework built upon pseudo-labeling, and A$^2$Net Xia et al. (2021) is an adversarial SFDA framework. Additionally, AaD Yang et al. (2022) features a more advanced contrastive learning objective tailored for solving SFDA problems, while NRC Yang et al. (2021a) utilizes a hierarchical neighborhood searching strategy.

The results, presented in Table 10, clearly indicate that our proposed SiLAN augmentation significantly improves the performance of both AaD and NRC. To be specific, our proposed SiLAN enhances the performance of HCL, A$^2$Net, NRC, and AaD by 0.6%, 2.0%, 1.2%, and 2.6%, respectively. These results demonstrate that SiLAN is an effective latent augmentation method for addressing SFDA problems across various SFDA frameworks.

Table 10: Ablation studies of integrating SiLAN into various SFDA frameworks for enhanced performance on VisDA2017 (ResNet-101).

| Method | plane | bcycl | bus | car | horse | knife | mcycl | person | plant | sktbrd | train | truck | Avg. |
|---|---|---|---|---|---|---|---|---|---|---|---|---|---|
| ResNet-101 He et al. (2016) | 55.1 | 53.3 | 61.9 | 59.1 | 80.6 | 17.9 | 79.7 | 31.2 | 81.0 | 26.5 | 73.5 | 8.5 | 52.4 |
| HCL Huang et al. (2021a) | 93.3 | 85.4 | 80.7 | 68.5 | 91.0 | 88.1 | 86.0 | 78.6 | 86.6 | 88.8 | 80.0 | 74.7 | 83.5 |
| A$^2$Net Xia et al. (2021) | 94.0 | 87.8 | 85.6 | 66.8 | 93.7 | 95.1 | 85.8 | 81.2 | 91.6 | 88.2 | 86.5 | 56.0 | 84.3 |
| NRC Yang et al. (2021a) | 96.1 | 90.8 | 83.9 | 61.5 | 95.7 | 95.7 | 84.4 | 80.7 | 94.0 | 91.9 | 89.0 | 59.5 | 85.3 |
| AaD Yang et al. (2022) | 95.2 | 90.5 | 85.5 | 79.2 | 96.4 | 96.2 | 88.8 | 80.4 | 93.9 | 91.8 | 91.1 | 55.9 | 87.1 |
| **SiLAN (Ours)** | 97.5 | 90.1 | 85.8 | 80.4 | **97.6** | 95.5 | 92.0 | 82.9 | **96.5** | **95.3** | 92.6 | 53.4 | 88.3 |
| **HCL+SiLAN (Ours)** | 94.6 | 85.4 | 83.2 | 67.3 | 94.2 | 86.5 | 86.3 | 80.8 | 88.2 | 85.2 | 83.4 | 74.6 | 84.1 |
| **A$^2$Net+SiLAN (Ours)** | 97.2 | 91.2 | **87.4** | 66.8 | 96.9 | **96.9** | 88.0 | 80.9 | 93.5 | 93.3 | 91.1 | 53.2 | 86.3 |
| **NRC+SiLAN (Ours)** | 97.2 | 90.5 | 84.3 | 63.8 | 96.7 | 95.4 | 86.2 | 85.1 | 95.6 | 93.2 | 90.2 | 59.8 | 86.5 |
| **AaD+SiLAN (Ours)** | **98.4** | **91.8** | 86.2 | **80.6** | 96.3 | 95.7 | **94.4** | **87.5** | 95.8 | 94.2 | **93.4** | 62.1 | **89.7** |

## A.8 Sensitivity Analysis on The Effect of The Source Pre-Training

In this sensitivity analysis, we evaluate how the quality of the source pre-trained model impacts target classification performance. We select source pre-trained models based on their performance on the source dataset's test sets, consistent with common practice in other SFDA methods.

To validate this approach, we conduct sensitivity analysis on the mid-scale Office-Home dataset using ResNet-50. We maintain consistency by using the same target dataset (*Artistic*) while varying the source domain datasets (*Clipart*, *Product*, and *Real-World*). Figure 5 illustrates the experimental results of the sensitivity analysis conducted on the effect of the source model pretraining on the target-domain classification performance. The left subfigure of Figure 5 illustrates the relationship between test performance on the source test set and the number of epochs used to pre-train the model on the source dataset. On the other hand, the right subfigure demonstrates how the quality of the source pre-trained model, measured by the number

of epochs used for pre-training on the source dataset, influences the classification performance in the target domain after adaptation convergence. The results indicate that, in the context of our SiLAN approach, selecting models that exhibit superior performance on the source test set as the source pre-trained model generally leads to optimal target adaptation outcomes.

The diversity of the source dataset also influences target adaptation performance. For instance, the synthetic (source) domain in the VisDA-2017 dataset includes 152,397 images generated from 3D models across 12 object categories. These images feature diverse shapes, colors, textures, and sizes. In contrast, real (target) domain images vary in backgrounds, lighting, occlusions, and object poses. This diversity poses challenges for domain adaptation algorithms, requiring effective generalization across disparities for optimal target domain performance.

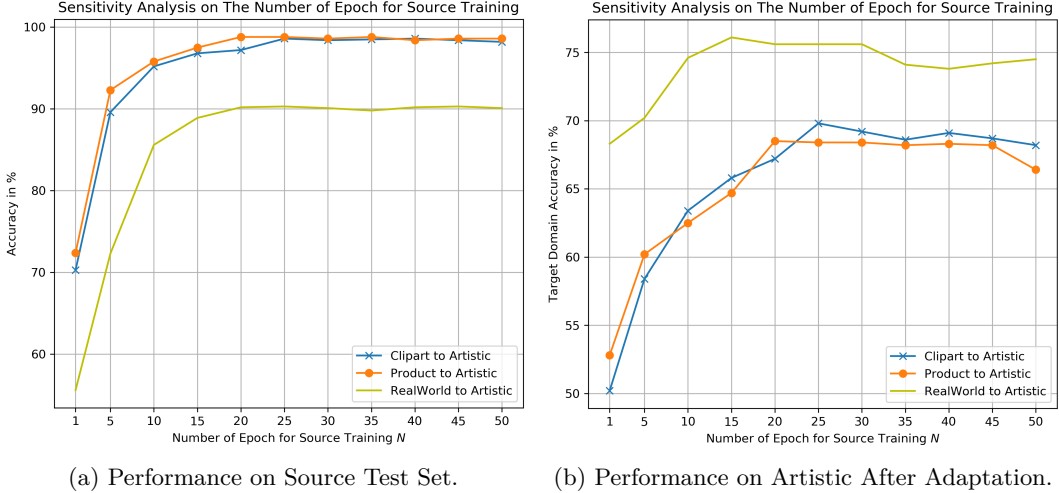

(a) Performance on Source Test Set.      (b) Performance on Artistic After Adaptation.

Figure 5: Sensitivity analysis on the Office-Home dataset using ResNet-50 to evaluate how the quality of the source pre-trained model impacts target-domain classification performance.

## A.9 Additional Experiments on Vision Transformers

To demonstrate the effectiveness of our SiLAN across different backbone architectures, we performed experiments on all benchmark datasets using the ViT-16-B vision transformer as our backbone, as suggested in prior work Xu et al. (2021). The results, presented in Tables 11, 12, and 13, highlight the efficacy of our SiLAN in addressing SFDA problems with a vision transformer backbone, outperforming existing ViT-based SFDA methods.

Table 11: Comparison of SFDA methods using ViT-B-16 on *Office-31*.

| Method | A→D | A→W | D→W | D→A | W→D | W→A | Avg. |
|---|---|---|---|---|---|---|---|
| ViT-B-16 | 90.8 | 90.4 | 76.8 | 98.2 | 76.4 | 100.0 | 88.8 |
| TVT Yang et al. (2023) | 96.4 | 96.4 | 84.9 | 99.4 | 86.1 | 100.0 | 93.8 |
| CDTrans Xu et al. (2021) | 97.0 | 96.7 | 81.1 | 99.0 | 81.9 | 100.0 | 92.6 |
| **SiLAN (Ours)** | 95.3 | 97.2 | **88.1** | **99.6** | **89.2** | 100.0 | **94.6** |
| **CDTrans+SiLAN (Ours)** | **97.4** | **98.1** | 83.4 | 99.4 | 83.3 | **100.0** | 93.6 |

Table 12: Comparison of the SFDA methods on *Office-Home* (ViT-B-16).

| Method | Ar→Cl | Ar→Pr | Ar→Rw | Cl→Ar | Cl→Pr | Cl→Rw | Pr→Ar | Pr→Cl | Pr→Rw | Rw→Ar | Rw→Cl | Rw→Pr | Avg. |
|---|---|---|---|---|---|---|---|---|---|---|---|---|---|
| ViT-B-16 | 61.8 | 79.5 | 84.3 | 75.4 | 78.8 | 81.2 | 72.8 | 55.7 | 84.4 | 78.3 | 59.3 | 86.0 | 74.8 |
| TVT Yang et al. (2023) | **74.9** | 86.8 | **89.5** | 82.8 | 88.0 | **88.3** | 79.8 | 71.9 | **90.1** | 85.5 | 74.6 | 90.6 | 83.6 |
| CDTrans Xu et al. (2021) | 68.8 | 85.0 | 86.9 | 81.5 | 87.1 | 87.3 | 79.6 | 63.3 | 88.2 | 82.0 | 66.0 | 90.6 | 80.5 |
| **SiLAN (Ours)** | 70.4 | **90.5** | 88.6 | **85.3** | 83.1 | 86.5 | **85.2** | 73.9 | 88.6 | **86.1** | **80.0** | **92.5** | **84.2** |
| **CDTrans+SiLAN (Ours)** | 72.1 | 86.2 | 85.4 | 81.7 | **88.5** | 86.9 | 82.4 | **81.1** | 87.9 | 84.5 | 78.2 | 90.4 | 83.8 |

Table 13: Comparison of the SFDA methods on *VisDA2017* (ViT-B-16).

| Method | plane | bcycl | bus | car | horse | knife | mcycl | person | plant | sktbrd | train | truck | Avg. |
|---|---|---|---|---|---|---|---|---|---|---|---|---|---|
| ViT-B-16 | **97.7** | 48.1 | 86.6 | 61.6 | 78.1 | 63.4 | 94.7 | 10.3 | 87.7 | 47.7 | **94.4** | 35.5 | 67.1 |
| TVT Yang et al. (2023) | 92.9 | 85.6 | 77.5 | 60.5 | 93.6 | 98.2 | 89.4 | 76.4 | 93.6 | 92.0 | 91.7 | 55.7 | 83.9 |
| CDTrans Xu et al. (2021) | 97.1 | 90.5 | 82.4 | **77.5** | 96.6 | 96.1 | 93.6 | 88.6 | 97.9 | 86.9 | 90.3 | 62.8 | 88.4 |
| **SiLAN (Ours)** | 92.5 | 90.1 | **92.4** | 70.6 | 92.1 | **98.5** | 95.8 | 89.2 | 94.5 | **93.3** | 90.6 | **64.4** | 88.7 |
| **CDTrans+SiLAN (Ours)** | 96.8 | **92.5** | 86.2 | 75.2 | **98.5** | 95.5 | **95.8** | 90.6 | **98.2** | 92.1 | 87.5 | 63.2 | **89.3** |

# B Proof

## B.1 Proof of Proposition 1

**Proposition 1.** The InfoNCE-based contrastive loss, denoted as $\mathcal{L}_{cont}$, serves as an upper bound for achieving two distinct alignments in the output logit space. Formally,

$$\sum_{i=1}^{m}\left(log(m-1) + \frac{||f_t(\mathbf{x}_i) - f_t(\mathbf{x}_i^+)||_2^2}{2\tau} + \frac{1}{m-1}\sum_{j\neq i} -\frac{||f_t(\mathbf{x}_i) - f_t(\mathbf{x}_j^+)||_2^2}{2\tau}\right) \leq \mathcal{L}_{cont},$$

where with $\tau$ being a temperature and $m$ being the size of a mini-batch, $\mathcal{L}_{cont}$ is defined as,

$$\mathcal{L}_{cont} = -\sum_{i=1}^{m} log \frac{e^{f_t^\top(\mathbf{x}_i)f_t(\mathbf{x}_i^+)/\tau}}{\sum_{j\neq i} e^{f_t^\top(\mathbf{x}_i)f_t(\mathbf{x}_j^+)/\tau}}.$$

*Proof.* the InfoNCE loss defined in the output logit space is formulated as:

$$\begin{aligned} \mathcal{L}_{cont} &= -\sum_{i=1}^{m} log \frac{e^{f_t^\top(\mathbf{x}_i)f_t(\mathbf{x}_i^+)/\tau}}{\sum_{j\neq i} e^{f_t^\top(\mathbf{x}_i)f_t(\mathbf{x}_j^+)/\tau}} \\ &= \sum_{i=1}^{m}\left(-f_t^\top(\mathbf{x}_i)f_t(\mathbf{x}_i^+)/\tau + log\left(\frac{m-1}{m-1}\sum_{j\neq i} e^{f_t^\top(\mathbf{x}_i)f_t(\mathbf{x}_j^+)/\tau}\right)\right). \end{aligned} \tag{6}$$

Then, we apply Jensen's inequality to Equation 6:

$$\begin{aligned} \mathcal{L}_{cont} &\geq \sum_{i=1}^{m}\left(-f_t^\top(\mathbf{x}_i)f_t(\mathbf{x}_i^+)/\tau + \frac{1}{m-1}\sum_{j\neq i} log\left((m-1)\, e^{f_t^\top(\mathbf{x}_i)f_t(\mathbf{x}_j^+)/\tau}\right)\right) \\ &= \sum_{i=1}^{m}\left(-f_t^\top(\mathbf{x}_i)f_t(\mathbf{x}_i^+)/\tau + \frac{1}{m-1}\sum_{j\neq i} f_t^\top(\mathbf{x}_i)f_t(\mathbf{x}_j^+)/\tau + \frac{1}{m-1}\sum_{j\neq i} log(m-1)\right) \\ &= \sum_{i=1}^{m}\left(log(m-1) - f_t^\top(\mathbf{x}_i)f_t(\mathbf{x}_i^+)/\tau + \frac{1}{m-1}\sum_{j\neq i} f_t^\top(\mathbf{x}_i)f_t(\mathbf{x}_j^+)/\tau\right) \end{aligned} \tag{7}$$

Then, express cosine similarity between two functions in terms of Euclidean distance:

$$-u^\top v := \frac{||u - v||_2^2}{2} - 1. \tag{8}$$

Plugging Eqn 8 into Inequality 7:

$$\sum_{i=1}^{m} \left( log(m-1) + \frac{||f_t(\mathbf{x}_i) - f_t(\mathbf{x}_i^+)||_2^2}{2\tau} + \frac{1}{m-1} \sum_{j \neq i} -\frac{||f_t(\mathbf{x}_i) - f_t(\mathbf{x}_j^+)||_2^2}{2\tau} \right) \leq \mathcal{L}_{cont}, \tag{9}$$

End of the proof. $\square$

## B.2 Proof of Lemma 2

**Lemma 2.** If $\mathcal{C}_z^\delta \cap \mathcal{C}_l^\delta = \emptyset$ holds for any $l \neq z$, then the error $\epsilon_{\mathcal{D}_T}$ defined on the groups of logits is upper bounded by:

$$\epsilon_{\mathcal{D}_T} \leq R_\delta + \sum_{z=1}^{Z} (\mathbb{P}[f_t(\mathbf{x}^+) \neq \mathbf{y}_t, \forall \mathbf{x}^+ \in \mathcal{S}_z^+] + \mathbb{P}[f_t(\mathbf{x}^-) = \mathbf{y}_t, \forall \mathbf{x}^- \in \mathcal{S}_z^-]),$$

where $R_\delta = \frac{\cup_{z=1}^{Z}(\mathcal{C}_z - \mathcal{C}_z^\delta)}{\cup_{z=1}^{Z}\mathcal{C}_z}$ and $(\mathcal{C}_z - \mathcal{C}_z^\delta)$ may overlap with $(\mathcal{C}_l - \mathcal{C}_l^\delta)$ for any $l \neq z$.

*Proof.* Based on Huang et al. (2021b), our study refines the theoretical problem of target classification error within the context of contrastive clustering. It decomposes the classification error, focusing on the groups of output logits $\mathcal{C}_z$ where $z \in [1, Z]$, as illustrated in Equation 3.

The first error term can only occur within the overlapping regions (*i.e.,* intersections) between the groups of logits generated through contrastive learning Huang et al. (2021b). Therefore, we have:

$$\sum_{z=1}^{Z} \mathbb{P}[f_t(\mathbf{x}_t) \neq z, \forall \mathbf{x}_t \in \mathcal{C}_z] \leq \mathbb{P}[\overline{\cup_{z=1}^{Z}\mathcal{C}_z^\delta}]. \tag{10}$$

The informativeness of the group assignment depends entirely on the informativeness of the transformations of queries when optimizing the contrastive loss. Thus, the upper bound of the second error term can be established as:

$$\sum_{z=1}^{Z} \mathbb{P}[z \neq \mathbf{y}_t, \forall \mathbf{x}_t \in \mathcal{C}_z] \leq \sum_{z=1}^{Z} \left( \mathbb{P}[f_t(\mathbf{x}^+) \neq \mathbf{y}_t, \forall \mathbf{x}^+ \in \mathcal{S}_z^+] + \mathbb{P}[f_t(\mathbf{x}^-) = \mathbf{y}_t, \forall \mathbf{x}^- \in \mathcal{S}_z^-] \right), \tag{11}$$

where the first term represents the error of assigning the logits of positive samples to a group that does not correspond to the ground truth of the query; and the second term denotes the error that occurs when assigning the logits of negative samples to the group to which the query should belong based on its ground truth.

Incorporating Inequalities 10 and 11 into Equation 3, we have:

$$\epsilon_{\mathcal{D}_T} \leq \mathbb{P}[\overline{\cup_{z=1}^{Z}\mathcal{C}_z^\delta}] + \sum_{z=1}^{Z} \left( \mathbb{P}[f_t(\mathbf{x}^+) \neq \mathbf{y}_t, \forall \mathbf{x}^+ \in \mathcal{S}_z^+] + \mathbb{P}[f_t(\mathbf{x}^-) = \mathbf{y}_t, \forall \mathbf{x}^- \in \mathcal{S}_z^-] \right)$$

$$= 1 - \frac{\cup_{z=1}^{Z}\mathcal{C}_z^\delta}{\cup_{z=1}^{Z}\mathcal{C}_z} + \sum_{z=1}^{Z} \left( \mathbb{P}[f_t(\mathbf{x}^+) \neq \mathbf{y}_t, \forall \mathbf{x}^+ \in \mathcal{S}_z^+] + \mathbb{P}[f_t(\mathbf{x}^-) = \mathbf{y}_t, \forall \mathbf{x}^- \in \mathcal{S}_z^-] \right).$$

End of the proof. $\square$

### B.3 Proof of Proposition 3

**Proposition 3.** If $\mathcal{C}_z^\delta \cap \mathcal{C}_l^\delta = \emptyset$ holds for any $z \neq l$, and the assumption that the representation of query samples in the feature space will stay close to their positive augmentations and be far away from their negative samples holds, then $\forall z \neq l$ and $i \neq j$ (where $\mathbf{x}_i \in \mathcal{C}_z^\delta$ and $\mathbf{x}_j \in \mathcal{C}_l^\delta$), their distance in the latent space is lower bounded when we take the limit $\sigma_{ext}^2 \to 0$ as follows:

$$|G_t^{opt}(\mathbf{x}_i) - G_t^{opt}(\mathbf{x}_j)| \geq 3.1704\sigma,$$

*Proof.* Inspired by the methodology introduced in Hogg & Lang (2013) for calculating the effective region of a Gaussian beam, we introduce the concept of the Gaussian beam to estimate the effective region of our SiLAN augmentation, whose effect is within a Gaussian profile. Here, all the possible augmented views, originating from a query sample $\mathbf{x}$, center at the mean $\mu_K(\mathbf{x})$ of the neighborhood of the query.

Hence, a radial profile with radius $R$ and centroid $\mu_K(\mathbf{x})$ can be employed to represent the effective region of the potential augmented views from SiLAN:

$$AR(R) = \int_0^R \hat{\mathbf{h}}(r)2\pi r dr,$$

where $\hat{\mathbf{h}}(r)$ is a radial profile function well-sampled and determined by SiLAN and the feature extractor parameters.

Assuming perfect alignment is achieved after $G_t^{opt}$ converges on the contrastive objective, meaning positive augmentations (keys) lie in the same cluster as the query while negative keys do not lie in the query's cluster, then, the gap between any two distinct clusters will be determined by the augmentation that belongs to one cluster and is nearest to the other cluster. In summary, under perfect alignment, there is no overlap between the Gaussian profiles of samples from different clusters. Thus, we have:

$$|G_t^{opt}(\mathbf{x}_i) - G_t^{opt}(\mathbf{x}_j)| \geq 2R.$$

Then, considering the extraneous noise with variance $\sigma_{ext}^2$, the total variance in the informative radius, with respect to $\mu_k(\mathbf{x})$, out to $R$ can be written as:

$$\sigma_{total}^2(R) = AR(R) + \pi R^2 \sigma_{ext}^2,$$

where $\pi R^2$ is the aperture area given $R$, containing possible augmented views centered at $\mu_k(\mathbf{x})$.

The transformation-to-noise ratio $T/N$, which is similar to the definition of signal-to-noise ratio to quantify Gaussian beam Hogg & Lang (2013), as a function of radius $R$ can be written as:

$$T/N = \frac{AR(R)}{\sqrt{AR(R) + \pi R^2 \sigma_{ext}^2}}$$
$$= \frac{\int_0^R \hat{\mathbf{h}}(r)2\pi r dr}{\sqrt{\int_0^R \hat{\mathbf{h}}(r)2\pi r dr + \pi R^2 \sigma_{ext}^2}}.$$

After adding random noise $\xi$, we have a 2D Gaussian for the source profile centered at $\mu_K(\mathbf{x})$:

$$\hat{\mathbf{h}}(r) = \frac{1}{2\pi\sigma^2} e^{\frac{-r^2}{2\sigma^2}},$$

where $\sigma$ is the standard deviation of the profile. Then,

$$T/N = \frac{1 - e^{\frac{-R^2}{2\sigma^2}}}{\sqrt{1 - e^{\frac{-R^2}{2\sigma^2}} + \pi R^2 \sigma_{ext}^2}}.$$

Taking $\frac{\partial (T/N)}{\partial R} = 0$ and the limit as $\sigma_{ext}^2 \to 0$, we have the optimum radius $R \approx 1.5852\sigma$.

By incorporating $R \approx 1.5852\sigma$ into the inequality, we have:

$$|G_t^{opt}(\mathbf{x}_i) - G_t^{opt}(\mathbf{x}_j)| \geq 3.1704\sigma,$$

End of the proof. □

### B.4   Proof of Lemma 4

**Lemma 4.**   $\forall z \neq l$ and $i \neq j$, if the linear classifier $F^{opt}$ is L-bi-Lipschitz continuous, $\mathcal{C}_z^\delta \cap \mathcal{C}_l^\delta = \emptyset$ holds for any $z \neq l$ where $\mathbf{x}_i \in \mathcal{C}_z^\delta$ and $\mathbf{x}_j \in \mathcal{C}_l^\delta$, and $\sigma_{ext}^2 \to 0$:

$$|f_t^{opt}(\mathbf{x}_i) - f_t^{opt}(\mathbf{x}_j)| \geq \frac{3.1704\sigma}{L},$$

where $f_t^{opt} = F_t^{opt}(G_t^{opt}(\cdot))$.

*Proof.* As the classifier $F$ is linear and $L$-bi-Lipschitz continuous, we have:

$$
\begin{aligned}
|f_t^{opt}(\mathbf{x}_i) - f_t^{opt}(\mathbf{x}_j)| &= |F_t^{opt}(G_t^{opt}(\mathbf{x}_i)) - F_t^{opt}(G_t^{opt}(\mathbf{x}_j))| \\
&\geq \frac{1}{L}|G_t^{opt}(\mathbf{x}_i) - G_t^{opt}(\mathbf{x}_j)|.
\end{aligned}
$$

From Proposition 3, we have $|G_t^{opt}(\mathbf{x}_i) - G_t^{opt}(\mathbf{x}_j)| \geq 3.1407\sigma$, thus

$$|f_t^{opt}(\mathbf{x}_i) - f_t^{opt}(\mathbf{x}_j)| \geq \frac{1}{L}3.1407\sigma$$

End of the proof. □

## C   Further Intuitive Insights into The Theoretical Work

In this section, we aim to provide practitioners with a clearer understanding of our proposed SiLAN. To achieve this, we offer more intuitive explanations and visual illustrations to clarify the purpose and outcomes of our theoretical work.

### C.1   Intuition of Proposition 3

Proposition 3 establishes a theoretical lower bound for the L1 distance between the latent features of data samples belonging to different logit clusters. This bound is derived under the assumption that the classification model is optimized by a contrastive loss to align the query predictions with those of our SiLAN augmentations, while intentionally misaligning the query predictions with those of the data samples within the same mini-batch.

This analysis presupposes that perfect alignment occurs after $G_t^{opt}$ converges on such a contrastive objective. In other words, positive augmentations (keys) are positioned within the same cluster as the query, while negative keys do not lie within the query's cluster. As illustrated in Figure 6, the gap between any two distinct clusters (i.e., $|G_t^{opt}(\mathbf{x}_i) - G_t^{opt}(\mathbf{x}_j)|$ where $\mathbf{x}_i$ and $\mathbf{x}_j$ belong to different logit clusters) is determined by the augmentation nearest to one cluster but belonging to the other.

It is important to recall that our SiLAN augmentation introduces Gaussian noise, incorporating guidance from the $k$NNs found by the source pre-trained model to the centroid of the query neighborhood found by

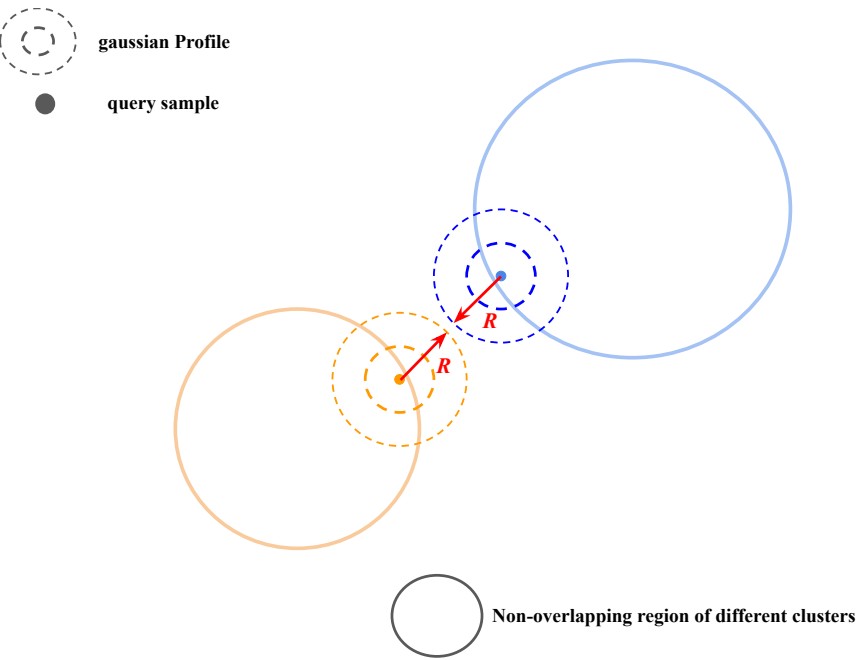

Figure 6: The minimum gap between the non-overlapping regions of two clusters has a lower bound determined by the radius of the Gaussian profile.

the current target model. Mathematically, the augmented features used to generate the positive keys for contrastive clustering can be represented as:

$$\hat{\mathbf{h}} := G_t(\mu_K^t(\mathbf{x_t}^i)) + \xi. \tag{12}$$

Here, $\mathbf{x_t}^i$ denotes the target query sample, while $\mu_K^t(\mathbf{x_t}^i)$ denotes the centroid of its $k$NN neighborhood. The noise $\xi$ for the augmentation (extraneous noise introduced in Proposition 3), drawn from a Gaussian distribution $\mathcal{N}(0, \sigma_K^s{}^2(\mathbf{x_t}^i))$, is determined by the variance of the kNNs as determined by the source pre-trained model.

Thus, employing our SiLAN-augmented neighbors as positive keys for contrastive clustering enables the establishment of a lower bound on the distance between logit clusters, as our SiLAN induces a Gaussian profile for the positive augmentation. Proposition 3 utilizes a geometric approach Hogg & Lang (2013) to calculate the effective region of a Gaussian beam (in our case, the centroid of the query's target neighbors serves as the center of this Gaussian beam) in computational optics to determine this lower bound.

## C.2   Intuition of Lemma 4

Proposition 3 establishes the theoretical lower bound of the L1 distance between latent features belonging to different logit clusters. However, our primary interest lies not in the cluster distance within the latent feature space, but rather in the distance lower bound between the output logits of different clusters. A larger distance lower bound in the output logit space facilitates the derivation of decision boundaries for target classification.

This distance lower bound in the latent feature space can be straightforwardly extended to the output logit space if the classifier is linear. Hence, we employ a linear fully-connected classifier in all our experiments to ensure consistency between our theoretical framework and empirical implementations.

### C.3 Derivation Details of Equation 3

In this section, we provide additional details regarding the derivation of the error term represented by Equation 3, which forms the cornerstone of our theoretical analysis.

To initiate our analysis, we first reinterpret the classification error on the target domain $\epsilon_{D_T} = \frac{number\ of\ misclassified\ samples}{total\ number\ of\ samples}$ by considering it as the probability of the model's predictions failing to align with the true labels of corresponding samples, given the current target classification model $f_t$. The classification error quantifies the ratio of the misclassified instances to the total number of instances. Mathematically, this is represented as:

$$\epsilon_{D_T} = \frac{number\ of\ misclassified\ samples}{total\ number\ of\ samples} = \frac{\sum_{i=1}^{N}(I(f_t(x^i) \neq y^i]))}{N} = P[f_t(X_T) \neq Y_T].$$

Here, $I$ is the binary indictor that equals one when $f_t(x^i) \neq y^i$.

Instead of evaluating misalignment based solely on individual sample indices, we refine this analysis to focus on samples within each cluster $z$ (where the number of clusters corresponds to the number of neurons in the last fully connected layer of the classifier). Thus, we express the classification error as:

$$\epsilon_{D_T} = \sum_{z=1}^{Z}(P_z(f_t(X_T^z \neq Y_T^z))).$$

Here, $X_T^z$ and $Y_T^z$ indicate the target samples and their labels with cluster $z$. Therefore, after model convergence and cluster formation, errors or misalignments can occur due to two main factors:

- Incorrect cluster assignment, where the model assigns a sample to the wrong cluster ($P[z \neq y_t], \forall x_t \in C_z$).

- Overlapping between clusters leading to ambiguous model predictions for samples within cluster $z$ ($P[f_t(x_t \neq z)], \forall x_t \in C_z$).

By re-examining the classification error in the context of possible errors during the cluster formation process as Equation 3, we can then proceed with the subsequent analysis to investigate how contrastive clustering influences the performance of target classification.

### C.4 Interconnections Among Three Theoretical Insights on Contrastive SFDA

In Section 4.2, we identified three overlooked factors in existing contrastive SFDA methods based on our theoretical analysis. Here, we clarify the interconnections among these insights and elucidate how our exploration of them informs our proposed solution to enhance the contrastive SFDA framework.

In summary, examining Insights 1 and 2 provides the foundational understanding necessary to address Insight 3 effectively. Insight 1 highlights the inadequacy of relying solely on data augmentation for generating positive keys in contrastive SFDA. This realization motivates us to explore latent augmentation as an alternative way for positive key generation. Expanding on Insight 1, our empirical observations from latent t-SNE analysis shown in Figure 1 indicate that generating target features through the source pre-trained model, which also initializes the target adaptation model, provides valuable information for target-domain classification beyond mere initialization. This information, as highlighted in the two observations in the introduction, indicates that neighboring target data points, determined in the latent feature space by the source pre-trained model, still share the same labels. As a result, we explore Insight 2 to examine whether we can capture this crucial information by controlling the range of neighborhood searching. To be specific, we investigate how the number $k$ of $k$NN influences contrastive SFDA.

Insight 3 highlights an overlooked problem, which can be addressed through a straightforward solution derived from the analyses of Insights 1 and 2: incorporating guidance from the $k$NNs identified in the source-pre-trained model-determined latent feature space directly into the latent augmentation process to generate the positive key for contrastive clustering given a query sample. Our subsequent studies in Sections 5.4 and 5.5 demonstrate the significance of the standard deviation of the Gaussian noise for such latent augmentation, as it determines the range of neighborhood searching. Thus, by utilizing the standard deviation derived

from the $k$NNs of the source pre-trained model to control the range of latent Gaussian augmentation, we enable the model update to identify neighborhoods in the target domain that exhibit consistent ground truth alignment with the current target query sample. In simpler terms, employing positive keys generated in this manner steers the clustering process toward samples sharing the same ground truth. In essence, our derivations of Insights 1, 2, and 3 are interrelated and complementary, leading us to our final solution.

To sum up, addressing insight 3 involves setting the standard deviation of the Gaussian used to generate positive keys in the latent space (*Insight 1*) for contrastive clustering to match the standard deviation of the neighborhood determined by the source pre-trained model (*Insight 2*). This neighborhood comprises samples sharing the same ground truth as the query target sample. Conducting latent augmentation based on this profile is crucial, as it offers additional guidance regarding the target ground truth to contrastive clustering.

