# OpenReview forum: "What Has Been Overlooked in Contrastive Source-Free Domain Adaptation: Leveraging Source-Informed Latent Augmentation within Neighborhood Context"
_TMLR — Accepted by TMLR_

### Review · Reviewer_rFpU · 2024-02-29

**Summary Of Contributions:**

This paper investigates the topic of source-free domain adaptation and discusses a classic contrastive learning approach.

(1) The author first provides a theoretical analysis of SFDA based on contrastive learning and then derives several insights from the theoretical analysis.

(2) Inspired by these insights, the author proposes a method (SiLAN) that combines augmentation and neighbor information methods, generating better positive key pairs by introducing information from the source model.

(3) Rigorous experimentation on various benchmarks substantiates that the proposed method attains state-of-the-art performance.

**Audience:**

Yes

**Claims And Evidence:**

Yes

**Requested Changes:**

Please refer to the weaknesses.

**Strengths And Weaknesses:**

Strengths:
1. The author combines augmentation with a neighbor-based approach and continues introducing information from the source model during training. This is quite innovative.
2. The performance is impressive, surpassing existing methods.

Weaknesses:
1. The cornerstone of this paper lies in formula (3), which raises the question of how the errors in the target domain can be adequately represented by these two terms. A comprehensive and detailed explanation is essential to elucidate this critical aspect.

2. The methodology section raises several concerns:
(a) The first insight posits that achieving domain adaptation without addressing distribution divergence is challenging. It remains ambiguous how this insight is derived from the theory proposed in this paper. Furthermore, the proposed method lacks a clear elucidation of how it mitigates domain divergence.
(b) The methodology section extensively delves into explaining the workings of contrastive learning and the theoretical errors related to contrastive learning in the target domain. This inclination towards unsupervised learning does not harmonize well with the theme of source-free domain adaptation.
(c) The first and second insights, being intuitive, do not necessarily require extensive theoretical analysis. The third insight, emphasizing the necessity of further leveraging source information, should be the primary focus of this paper. However, it remains unclear how the author addresses this crucial point.

3.  Several descriptions are ambiguous:
(a) In the fourth paragraph of the introduction, the author claims that various design choices for positive keys in constrive SFDA can mitigate performance degradation due to domain shift. However, it is unclear whether the author provides evidence for this statement in the paper. Existing SFDA works utilizing contrastive learning methods in SFDA do not seem to lead to performance degradation.
(b) In the introduction's summary, the author mentions "three often-overlooked factors," but these factors are not explicitly stated in the introduction, leading to confusion.

4.  Regarding the experiments:
(a) The paper lacks a comparison with the latest data augmentation-based method, SF(DA)^2, as presented by U. Hwang, et al. in their work "SF(DA)^2: Source-free Domain Adaptation Through the Lens of Data Augmentation," ICLR 2024.
(b) Considering the paper's approach, combining SiLAN with InfoNCE raises the question of whether similar analysis could be done with advanced SFDA methods like AaD or NRC.

---

> ### Author Response · Authors · 2024-03-19
> **Response to Reviewer rFpU (1)**
>
> We hugely thank the reviewer for acknowledging our work and providing valuable suggestions to improve our paper. In our response, we will address the identified weaknesses and incorporate the requested changes systematically, as outlined below:
>
> ***Weakness 1 (Detailed Explanation of Formula 3)***
>
> We would like to thank the reviewer for suggesting a thorough explanation of Equation 3, which forms the foundation of our theoretical analysis. To enhance the coherence of the derivation, we can refine the explanation by breaking it down into clearer steps and providing more explicit connections between the concepts introduced. Here is the revised derivation:
>
> To initiate our analysis, we first reinterpret the classification error on the target domain by considering it as the probability of the model's predictions failing to align with the true labels of corresponding samples, given the current classification model $f_t$. The classification error quantifies the ratio of misclassified instances to the total number of instances. Mathematically, this is represented as:
>
> $\epsilon_{D_T} = \frac{number of misclassified samples}{total number of samples} = \frac{\sum_{i=1}^{N} (I(f_t(x^{i}) \neq y^{i}]))}{N} = P[f_t(X_T) \neq Y_T]$.
>
> Here, $I$ is the binary indictor that equals one when $f_t(x^{i}) \neq y^{i}$.
>
> Instead of assessing misalignment based solely on individual sample indices, we refine this analysis to focus on samples within each cluster $z$ (where the number of clusters corresponds to the number of neurons in the last fully connected layer of the classifier). Thus, we express the classification error as:
>
> $\epsilon_{D_T} = \sum_{z=1}^{Z} (P_{z}(f_t(X_T^{z} \neq Y_T^{z})))$.
>
> Here, $X_T^{z}$ and $Y_T^{z}$ indicate the target samples and their labels with cluster $z$. Therefore, after model convergence and cluster formation, errors or misalignments can occur due to two main factors:
>
> 1. Incorrect cluster assignment, where the model assigns a sample to the wrong cluster ($P[z \neq y_t], \forall x_{t} \in C_{z}$).
>
> 2. Overlapping between clusters leading to ambiguous model predictions for samples within cluster $z$ ($P[f_{t}(x_{t} \neq z)], \forall x_{t} \in C_{z}$).
>
> By re-examining the classification error in the context of possible errors during the cluster formation process as Equation 3, we can then proceed with the subsequent analysis to investigate how contrastive clustering influences the performance of target classification.
>
> Due to the page limit, we have incorporated the detailed explanation above into a new section ***Appendix C.3 Derivation Details of Equation 3*** of the revised manuscript, indicating the revisions in blue for easy identification.

---

> ### Author Response · Authors · 2024-03-19
> **Response to Reviewer rFpU (2)**
>
> **Weakness 2 (Clarification on Methodology)**
>
> We appreciate the reviewer's questions regarding the three insights derived from our theoretical analysis. There seems to be some misunderstanding regarding these insights.
>
> ***(a) Clarification on First Insight***: Our first insight highlights that solely relying on augmentation-based contrastive learning frameworks without incorporating distribution alignment techniques may not effectively address SFDA problems. This insight does not focus on the challenge of achieving source-free domain adaptation without distribution alignment, but rather emphasizes that augmentation-based contrastive learning alone may not suffice for addressing SFDA. This conclusion stems from Lemma 2, which provides an upper bound indicating that augmentation-based contrastive clustering can only reduce the overlapping term $R_{\delta}$​ , while being unable to address other terms in Lemma 2. This limitation arises because standard augmentation methods inherently lack information regarding the target ground truth.
>
> Our proposed SiLAN is not aimed at directly quantifying the domain shift, which can be challenging in a source-free setting. Instead, our focus lies in enhancing the informativeness of positive key generation within contrastive clustering. This informativeness is derived from the source pre-trained model, as illustrated in ***Figure 1***. Despite the domain shift potentially affecting clustering performance negatively, neighboring target data points in the latent feature space often share the same ground truth. This neighborhood information remains crucial in guiding the clustering process within the unlabeled target domain. Therefore, we introduce a latent augmentation-based approach, leveraging the standard derivation of $k$-nearest neighbors provided by the source pre-trained model. This extra guidance, derived from the ground truth consistency within such neighborhoods, enhances the effectiveness of contrastive clustering.
>
> As suggested by the reviewer, to clarify, we have updated the first insight in the revised manuscript on page 7. The revised insight now states: "Standard data augmentation without distribution alignment in a contrastive SFDA framework may not effectively address misclassifications of positive keys due to insufficient information about the target ground truth." The changes have been highlighted in blue.
>
> ***(b) Clarification on Unsupervised Learning and Source-Free Domain Adaptation***: In source-free scenarios, conventional domain adaptation methods, especially those reliant on distribution alignment techniques, become impractical. This is because there is no source data available during target adaptation, making it impossible to conduct an empirical measure between the two data domains. Moreover, utilizing generative models to generate source-like data during target adaptation poses risks of source information leakage to clients, contrary to the primary goal of source-free domain adaptation, which prioritizes source data privacy while adapting the model to the target domain.
>
> These constraints characterize the target adaptation under SFDA settings as an unsupervised learning problem, where only a pre-trained model is available during target adaptation, and all data remain unlabeled. Consequently, our theoretical analysis, grounded in unsupervised learning and tailored specifically to source-free settings, aligns well with the overarching theme of addressing source-free domain adaptation challenges. Moreover, our research primarily focuses on enhancing contrastive-based SFDA frameworks, given their outstanding performance in addressing SFDA challenges. The theoretical analysis we propose offers valuable insights into the effectiveness of contrastive learning in SFDA and suggests potential avenues for further improvement.

---

> ### Author Response · Authors · 2024-03-19
> **Response to Reviewer rFpU (3)**
>
> ***(c) Clarifying How Exploration of Insights 1 and 2 Leads to the Solution of Insight 3***: We appreciate the reviewer for bringing up the confusion. In fact, insights 1 and 2 serve as foundational pillars for understanding how we address insight 3.
>
> Insight 1 highlights the inadequacy of relying solely on data augmentation for generating positive keys in contrastive SFDA. This realization motivates us to explore latent augmentation as an alternative way for positive key generation. Expanding on Insight 1, our empirical observations from latent t-SNE analysis (shown in ***Figure 1***) indicate that generating target features through the source pre-trained model, which also initializes the target adaptation model, provides valuable information for target-domain classification beyond mere initialization. This information, as highlighted in the two observations in the introduction, indicates that neighboring target data points, determined in the latent feature space by the source pre-trained model, still share the same labels. As a result, we explore Insight 2 to examine whether we can capture this crucial information by controlling the range of neighborhood searching. Specifically, we investigate how the number $k$ of $k$NN influences contrastive SFDA.
>
> Insight 3 highlights an overlooked problem, which can be addressed through a straightforward solution derived from the analyses of Insights 1 and 2: incorporating guidance from the $k$NNs identified in the source-pre-trained model-determined latent feature space directly into the latent augmentation process to generate the positive key for contrastive clustering given a query sample. Our subsequent studies in Sections 5.4 and 5.5 demonstrate the significance of the standard deviation of the Gaussian noise for such latent augmentation, as it determines the range of neighborhood searching. Thus, by utilizing the standard deviation derived from the $k$NNs of the source pre-trained model to control the range of latent Gaussian augmentation, we enable the model update to identify neighborhoods in the target domain that exhibit consistent ground truth alignment with the current target query sample. In simpler terms, employing positive keys generated in this manner steers the clustering process toward samples sharing the same ground truth. In essence, our derivations of insights 1, 2, and 3 are interrelated and complementary, leading us to our final solution.
>
> To sum up, addressing insight 3 involves setting the standard deviation of the Gaussian used to generate positive keys in the latent space (Insight 1) for contrastive clustering to match the standard deviation of the neighborhood determined by the source pre-trained model. This neighborhood comprises samples sharing the same ground truth as the query target sample (Insight 2). Conducting latent augmentation based on this profile is crucial, as it offers additional guidance regarding the target ground truth to contrastive clustering.
>
> In response to the reviewer's suggestion, we have included a new section titled ***Appendix C.4 Interconnections Among Three Theoretical Insights on Contrastive SFDA*** in the revised manuscript. This added section integrates the discussed insights to improve the clarity of our paper. The revised section is highlighted in blue for easy identification.

---

> ### Author Response · Authors · 2024-03-19
> **Response to Reviewer rFpU (4)**
>
> **Weakness 3 (Clarification on Ambious Descriptions)**
>
> We would like to thank the reviewer for raising the ambitious descriptions.
>
> ***(a)*** ***Providing Evidence for The First Sentence of Paragraph Four***: In the fourth paragraph of the introduction, we stated that "the present paper aims to offer a comprehensive insight into how various design choices for positive keys in contrastive SFDA can mitigate the performance degradation due to domain shift." This statement serves as a summary of the primary objective of our study, which is to address the challenge of domain adaptation, which is exactly the model performance degradation resulting from domain shift, in the source-free setting. It aims to provide clarity regarding the focus of our research rather than making any assertion requiring evidence.
>
> ***(a)*** ***Existing SFDA Methods Do Not Lead to Performance Degradation***: The main challenge in domain adaptation research is the mitigation of performance degradation resulting from domain shift. This degradation in performance occurs when applying a source pre-trained model directly to a new dataset (target domain) without any adaptation or fine-tuning, leading to significantly worse classification performance compared to what the model could achieve in the source domain. Therefore, the goal of every domain adaptation and SFDA method is to address this issue of performance degradation due to domain shift. It is important to note that we are not suggesting that existing SFDA methods inherently cause performance degradation, but rather emphasizing the common objective of these methods in mitigating the effects of domain shift on model performance.
>
> To prevent any potential confusion, as suggested by the reviewer, we have revised this sentence to provide further clarification: “The primary goal of our study is to conduct a thorough analysis of how the design of positive keys could impact target domain classification performance. Moreover, we seek to explore how insights obtained from this analysis can be utilized to enhance contrastive SFDA frameworks”.
>
> We have incorporated the revision into the revised manuscript and highlighted it in blue.
>
> ***(b) Insights Not Included in Introduction***: We thank the reviewer for highlighting this confusion. In response, we have revised the fifth paragraph of the introduction to explicitly mention the three often-overlooked factors, as suggested by the reviewer. Below, the reviewer can find the revised paragraph for reference. Additionally, in the revised manuscript, we have highlighted the revision in blue to facilitate easy identification.
>
> ***Revision in Paragraph 5***: Motivated by these observations, we conduct a comprehensive analysis of the underlying principles of contrastive SFDA and explore aspects that have been overlooked in the existing contrastive SFDA frameworks. To be specific, we identify three overlooked factors: 1) Standard data augmentation techniques might not effectively reduce the likelihood of the model misclassifying positive transformations; 2) increasing the number of nearest neighbors, i.e., larger $k$, results in smoother predictions but also leads to a greater overlap of logit clusters; and 3) effective utilization of the source pre-trained model to leverage neighborhood label consistency for enhancing the informativeness of positive key generation has not yet been explored.

---

> ### Author Response · Authors · 2024-03-19
> **Response to Reviewer rFpU (5)**
>
> **Weakness 4 (Suggested Experiments)**
>
> ***(a) Comparison with The Latest Baseline SF(DA)$^2$***: We appreciate the reviewer for suggesting the inclusion of the latest data augmentation-based method in our comparison. In response to the reviewer's suggestion, we have incorporated SF(DA)$^2$ as the baseline method for comparison across all benchmark datasets in our revised manuscript. For the VisDA-2017 and Office-31 datasets, we reported their results as reported in their original papers. As the results of the experiments on the Office-Home dataset are not reported in their paper, we reproduced them. The suggested comparison illustrates that our proposed SiLAN method is more effective than other augmentation-based SFDA methods. The revision has been incorporated into ***Tables 1, 2, and 3*** of the revised paper, with the suggested baseline highlighted in blue.
>
> ***Table. Comparison of the SFDA methods on Office-31 (ResNet-50)***.
>
> |  Method | A→D |A→W | D→W | D→A | W→D | W→A | Avg. |
> |:------------:|:--------:|:--------:|:--------:|:--------:|--------:|:--------:|:--------:|
> | ResNet-50 | 68.9 | 68.4 | 96.7 | 62.5 | 99.3 | 60.7 | 76.1 |
> | SHOT [10] | 94.0 | 90.1 | 98.4 | 74.7 | 99.9 | 74.3 | 88.6 |
> | NRC [5] | 96.0 | 90.8 | 99.0 | 75.3 | 100.0 | 75.0 | 89.4 |
> | 3C-GAN [12] | 92.7 | 93.7 | 98.5 | 75.3 | 99.8 | 77.8 | 89.6 |
> | HCL [3] | 94.7 | 92.5 | 98.2 | 75.9 | 100.0 | 77.7 | 89.8 |
> |***SF(DA)$^2$ [9]*** | 95.8 | 92.1 | 99.0 | 75.7 | 99.8 | 76.8 | 89.9 |
> |:------------:|:--------:|:--------:|:--------:|:--------:|--------:|:--------:|:--------:|
> | ***SiLAN (Ours)*** | 97.1 | 95.8 | 98.9 | 76.4 | 100.0 | 76.9 | 90.7 |
>
> ***Table. Comparison of the SFDA methods on Office-Home (ResNet-50).***
>
> |  Method | Ar→Cl | Ar→Pr | Ar→Rw |Cl→Ar | Cl→Pr | Cl→Rw | Pr→Ar | Pr→Cl | Pr→Rw | Rw→Ar | Rw→Cl | Rw→Pr | Avg. |
> |:------------:|:--------:|:--------:|:--------:|:--------:|--------:|:--------:|:--------:|:--------:|:--------:|:--------:|--------:|:--------:|:--------:|
> | ResNet-50 | 34.9 | 50.0 | 58.0 | 37.4 | 41.9 | 46.2 | 38.5 | 31.2 | 60.4 | 53.9 | 41.2 | 59.9 | 46.1 |
> | G-SFDA [11] | 57.9 | 78.6 | 81.0 | 66.7 | 77.2 | 77.2 | 65.6 | 56.0 | 82.2 | 72.0 | 57.8 | 83.4 | 71.3 |
> | SHOT [10] | 57.1 | 78.1 | 81.5 | 68.0 | 78.2 | 78.1 | 67.4 | 54.9 | 82.2 | 73.3 | 58.8 | 84.3 | 71.8 |
> | NRC [5] | 57.7 | 80.3 | 82.0 | 68.1 | 79.8 | 78.6 | 65.3 | 56.4 | 83.0 | 71.0 | 58.6 | 85.6 | 72.2 |
> | AaD [1] | 59.3 | 79.3 | 82.1 | 68.9 | 79.8 | 79.5 | 67.2 | 57.4 | 83.1 | 72.1 | 58.5 | 85.4 | 72.7 |
> | DaC [6]  | 59.5 | 79.5 | 81.2 | 69.3 | 78.9 | 79.2 | 67.4 | 56.4 | 82.4 | 74.0 | 61.4 | 84.4 | 72.8 |
> | ***SF(DA)$^2$ [9]*** | 57.8 | 80.2 | 81.5 | 69.5 | 79.2 | 79.4 | 66.5 | 57.2 | 82.1 | 73.3 | 60.2 | 83.8 | 72.6 |
> |:------------:|:--------:|:--------:|:--------:|:--------:|--------:|:--------:|:--------:|:--------:|:--------:|:--------:|--------:|:--------:|:--------:|
> | ***SiLAN (Ours)*** | 58.2 | 81.2 | 82.5 | 69.8 | 78.6 | 80.3 | 68.4 | 58.6 | 82.5 | 75.6 | 60.8 | 86.1 | 73.6
>
> ***Table. Comparison of the SFDA methods on VisDA-2017 (ResNet-101).***
>
> |  Method | plane |  bcycl | bus |car | horse | knife | mcycl | person | plant | sktbrd | train | truck | Avg |
> |:------------:|:--------:|:-------:|:--------:|:-------:|:--------:|:-------:|:--------:|:-------:|:--------:|:-------:|:--------:|:-------:|-------:|
> | ResNet-101 | 55.1 | 53.3 | 61.9 | 59.1 | 80.6 | 17.9 | 79.7 | 31.2 | 81.0 | 26.5 | 73.5 | 8.5 | 52.4 |
> | HCL [3] | 93.3 | 85.4 | 80.7 | 68.5 | 91.0 | 88.1 | 86.0 | 78.6 | 86.6 | 88.8 | 80.0 | 74.7 | 83.5 |
> |A^{2}Net [7] | 94.0 | 87.8 | 85.6 | 66.8 | 93.7 | 95.1 | 85.8 | 81.2 | 91.6 | 88.2 | 86.5 | 56.0 | 84.3 |
> | AaD [1] | 95.2 | 90.5 | 85.5 | 79.2 | 96.4 | 96.2 | 88.8 | 80.4 | 93.9 | 91.8 | 91.1 | 55.9 | 87.1|
> | NRC [5] | 96.1 | 90.8 | 83.9 | 61.5 | 95.7 | 95.7 | 84.4 | 80.7 | 94.0 | 91.9 | 89.0 | 59.5 | 85.3 |
> | ***SF(DA)$^2$ [9]*** | 96.8 | 89.3 | 82.9 | 81.4 | 96.8 | 95.7 | 90.4 | 81.3 | 95.5 | 93.7 | 88.5 | 64.7 | 88.1|
> |:------------:|:--------:|:-------:|:--------:|:-------:|:--------:|:-------:|:--------:|:-------:|:--------:|:-------:|:--------:|:-------:|-------:|
> |***SiLAN (Ours)*** | 97.5 | 90.1 | 85.8 | 80.4 | 97.6 | 95.5 | 92.0 | 82.9 | 96.5 | 95.3 | 92.6 | 53.4 | 88.3 |

---

> ### Author Response · Authors · 2024-03-19
> **Response to Reviewer rFpU (6)**
>
> ***(b) Combining Our SiLAN with Advanced SFDA Methods Like AaD and NRC***: We appreciate the reviewer's suggestion to conduct ablation studies integrating our SiLAN method into more advanced techniques such as AaD and NRC. Following the reviewers' advice, we conducted these studies on the large-scale VisDA-2017 dataset using ResNet-101. The results, presented in the table below, clearly indicate that our proposed SiLAN augmentation significantly improves the performance of both AaD and NRC by 2.6% and 1.2%, respectively. We have included the requested experiments in ***Appendix A.7 SiLAN as A General Latent Augmentation Method*** of the revised manuscript, denoted by highlighting the revisions in blue.
>
> The implementation details differ between AaD and NRC. Both methods utilize the predictions of query neighbors as positive keys to optimize the model. However, their distinction lies in the design of the loss function. AaD employs a contrastive objective tailored to source-free domain adaptation, while NRC utilizes a hierarchical neighborhood searching strategy. Our SiLAN method seamlessly integrates into their frameworks because both of them utilize $k$NN searching in the latent feature space. We maintain consistency by employing the same neighborhood searching method and settings (i.e., the same number of $k$) to locate neighbors using both the target model and the source pre-trained model. The mean and variance of the latent features are calculated separately for the neighbors obtained from the target model and the source pre-trained model. These mean and variance values then serve as parameters for Gaussian augmentation in the latent space, enabling the generation of predictions for the neighbors. We refrain from altering their original implementations, opting only to augment their neighbors in the latent space using SiLAN, without fine-tuning.
>
> ***Table. Ablation studies of integrating SiLAN into various SFDA frameworks for enhanced performance on VisDA2017 (ResNet-101).***
>
> |  Method | plane |  bcycl | bus |car | horse | knife | mcycl | person | plant | sktbrd | train | truck | Avg |
> |:------------:|:--------:|:-------:|:--------:|:-------:|:--------:|:-------:|:--------:|:-------:|:--------:|:-------:|:--------:|:-------:|-------:|
> | ResNet-101 | 55.1 | 53.3 | 61.9 | 59.1 | 80.6 | 17.9 | 79.7 | 31.2 | 81.0 | 26.5 | 73.5 | 8.5 | 52.4 |
> | HCL [3] | 93.3 | 85.4 | 80.7 | 68.5 | 91.0 | 88.1 | 86.0 | 78.6 | 86.6 | 88.8 | 80.0 | 74.7 | 83.5 |
> |A$^{2}$Net [7] | 94.0 | 87.8 | 85.6 | 66.8 | 93.7 | 95.1 | 85.8 | 81.2 | 91.6 | 88.2 | 86.5 | 56.0 | 84.3 |
> | AaD [1] | 95.2 | 90.5 | 85.5 | 79.2 | 96.4 | 96.2 | 88.8 | 80.4 | 93.9 | 91.8 | 91.1 | 55.9 | 87.1|
> | NRC [5] | 96.1 | 90.8 | 83.9 | 61.5 | 95.7 | 95.7 | 84.4 | 80.7 | 94.0 | 91.9 | 89.0 | 59.5 | 85.3 |
> |:------------:|:--------:|:-------:|:--------:|:-------:|:--------:|:-------:|:--------:|:-------:|:--------:|:-------:|:--------:|:-------:|-------:|
> |***SiLAN*** | 97.5 | 90.1 | 85.8 | 80.4 | 97.6 | 95.5 | 92.0 | 82.9 | 96.5 | 95.3 | 92.6 | 53.4 | 88.3 |
> |***HCL+SiLAN*** |  94.6 | 85.4 | 83.2 | 67.3 | 94.2 | 86.5 | 86.3 | 80.8 | 88.2 | 85.2 | 83.4 | 74.6 | 84.1 |
> |***A$^{2}$Net+SiLAN*** | 97.2 | 91.2 | 87.4 | 66.8 | 96.9 | 96.9 | 88.0 | 80.9 | 93.5 | 93.3 | 91.1 | 53.2 | 86.3 |
> |***NRC+SiLAN*** | 97.2 | 90.5 | 84.3 | 63.8 | 96.7 | 95.4 | 86.2 | 85.1 | 95.6 | 93.2 | 90.2 | 59.8 | 86.5 |
> |***AaD+SiLAN*** | 98.4 |91.8 | 86.2 | 80.6 | 96.3 | 95.7 | 94.4 | 87.5 | 95.8 | 94.2 | 93.4 | 62.1 | 89.7 |

---

> ### Author Response · Authors · 2024-03-19
> **Reference**
>
> [1] Shiqi Yang, Yaxing Wang, Kai Wang, Shangling Jui, et al. Attracting and dispersing: A simple approach for source-free domain adaptation. In Conference on Neural Information Processing Systems (NeurIPS), 2022.
>
> [2] Jian Liang, Dapeng Hu, and Jiashi Feng. Do we really need to access the source data? source hypothesis transfer for unsupervised domain adaptation. In International Conference on Machine Learning (ICML), pp. 6028–6039. PMLR, 2020.
>
> [3] Jiaxing Huang, Dayan Guan, Aoran Xiao, and Shijian Lu. Model adaptation: Historical contrastive learning for unsupervised domain adaptation without source data.
> Advances in Neural Information Processing Systems (NeurIPS), 34:3635–3649, 2021a.
>
> [4] Shiqi Yang, Yaxing Wang, Joost van de Weijer, Luis Herranz, and Shangling Jui.
> Generalized source-free domain adaptation. In Proceedings of the IEEE/CVF International Conference on Computer Vision (ICCV), pp. 8978–8987, 2021b.
>
> [5] Shiqi Yang, Joost van de Weijer, Luis Herranz, Shangling Jui, et al. Exploiting the intrinsic neighborhood structure for source-free domain adaptation. Advances in Neural Information Processing Systems, 34: 29393–29405, 2021a.
>
> [6] Ziyi Zhang, Weikai Chen, Hui Cheng, Zhen Li, Siyuan Li, Liang Lin, and Guanbin Li. Divide and contrast: Source-free domain adaptation via adaptive contrastive learning.
> In Conference on Neural Information Processing Systems (NeurIPS), 2022.
>
> [7] Xia, H., Zhao, H. and Ding, Z. Adaptive adversarial network for source-free domain adaptation. In Proceedings of the IEEE/CVF international conference on computer vision (CVPR), 2021.
>
> [8] Yang Liu, Wei Zhang, and June Wang, Source-free domain adaptation for semantic segmentation. In Proceedings of the IEEE/CVF Conference on Computer Vision and Pattern Recognition (CVPR), 2021.
>
> [9] Uiwon Hwang, Jonghyun Lee, Juhyeon Shin, and Sungroh Yoon. Sf(da)^2: Source-free domain adaptation through the lens of data augmentation. In The Twelfth International Conference on Learning Representations (ICLR), 2024.
>
> [10] Jian Liang, Dapeng Hu, and Jiashi Feng. Do we really need to access the source data? source hypothesis transfer for unsupervised domain adaptation. In International Conference on Machine Learning (ICML), pp. 6028–6039. PMLR, 2020.
>
> [11] Shiqi Yang, Yaxing Wang, Joost van de Weijer, Luis Herranz, and Shangling Jui. Generalized source-free domain adaptation. In Proceedings of the IEEE/CVF International Conference on Computer Vision (ICCV), pp. 8978–8987, 2021b.
>
> [12] Rui Li, Qianfen Jiao, Wenming Cao, Hau-San Wong, and Si Wu. Model adaptation: Unsupervised domain adaptation without source data. In Proceedings of the IEEE/CVF Conference on Computer Vision and Pattern Recognition (CVPR), pp. 9641–9650, 2020.

---

> ### Author Response · Authors · 2024-04-03
> **Prompt for Discussion: A Friendly Reminder**
>
> Dear Reviewer rFpU,
>
> We extend our sincere gratitude for your invaluable suggestions regarding our paper.
>
> We believe we have adequately addressed all your concerns in the responses provided above. However, should you have any further inquiries or require additional clarification on our responses, we are more than willing to engage in further discussion.
>
> Once again, we greatly appreciate your diligent efforts in reviewing our paper.
>
> Best regards,
>
> Authors 2099

---

### Review · Reviewer_Bpp9 · 2024-03-02

**Summary Of Contributions:**

The authors analyzed contrastive learning in SFDA and found that domain shift causes dispersion of target features, but neighboring features often still share the same class. This explains why contrastive clustering approaches can still be successful for SFDA.
They proposed a new latent augmentation method called SiLAN. This method uses the dispersion of latent features in the neighborhood of a query sample (as informed by the source model) to improve how positive keys are generated for contrastive learning.
The proposed method uses a simple InfoNCE contrastive loss, and yet outperforms prior SFDA approaches on standard benchmarks.

**Audience:**

Yes

**Broader Impact Concerns:**

Domain shift itself can introduce new biases or change how existing biases manifest. SFDA methods need careful analysis to understand how they interact with potentially biased datasets, especially in sensitive applications.

**Claims And Evidence:**

Yes

**Requested Changes:**

1. Report results in Table 1,2,3 using ViTs.

2. Add a table benchmarks trade-offs between performance gains and efficiency.

3. add more discussion on pretrained source model. How diverse the data it should be trained on?

**Strengths And Weaknesses:**

Strengths

Strong Theoretical Basis: In Section 4.2, The analysis of contrastive learning in SFDA provides a solid foundation for understanding how and why the proposed method works. The observation in Figure 1 about feature dispersion and neighborhood informativeness offers unique insights.

Novel and Effective Augmentation: The SiLAN method is a creative approach to address the issues identified in the analysis. It leverages domain knowledge without requiring direct access to the source domain data. Fig 2 and 3. well illustrated the idea.

SOTA Performance: Achieving state-of-the-art results on standard benchmarks demonstrates the effectiveness and practicality of the proposed method. The fact that good performance is achieved with a simple InfoNCE loss (Eq. 5) highlights the elegance of the approach.


Weaknesses

Dependency on Source Model: While the method doesn't use source data directly, it still relies on a pre-trained source model. Analyzing its sensitivity to the quality of this source model would be valuable.

Lack of Ablation Study: A thorough ablation study could help isolate the specific impact of the SiLAN augmentation and how it interacts with other components of the framework. Table 1, 2, 3 are based on ResNet, does it generalize well in ViTs?

Computational Overhead: It's worth investigating the computational cost of the neighborhood-based feature dispersion calculations and the potential trade-offs between performance gains and efficiency. I cannot find it anywhere in the paper.

---

> ### Author Response · Authors · 2024-03-19
> **Response to Reviewer Bpp9 (1)**
>
> We hugely thank the reviewer for acknowledging our work and providing valuable suggestions to improve our paper. In our response, we will address the identified weaknesses and incorporate the requested changes systematically, as outlined below:
>
> **Weakness 1 and Requested Changes 3 (Add More Discussion On Source Pre-Trained Model)**
>
> We would like to thank the reviewer for suggesting the analysis of how the target classification performance is influenced by the quality of the source pre-trained model. In our experiments, we selected the source pre-trained models based on their performance on the test sets of the source dataset. This approach aligns with the method commonly used by other SFDA techniques for selecting their source pre-trained models during the source pre-training phase.
>
> However, as suggested by the reviewer, it is important to validate our decision by examining how the sensitivity of the source pre-trained model affects the target-domain classification performance. To address this, we conducted the suggested sensitivity analysis on the mid-scale Office-Home dataset, utilizing the ResNet-50 backbone. We maintained consistency by conducting experiments using the same target dataset (Artistic) while varying the source domain datasets (Clipart, Product, and Real-World).
>
> ***Figure 5*** in ***Appendix A.8*** of the revised manuscript illustrates the experimental results of the sensitivity analysis conducted on the effect of the source model pretraining on the target-domain classification performance. The left subfigure of Figure 5 illustrates the relationship between test performance on the source test set and the number of epochs used to pre-train the model on the source dataset. On the other hand, the right subfigure demonstrates how the quality of the source pre-trained model, measured by the number of epochs used for pre-training on the source dataset, influences the classification performance in the target domain after adaptation convergence. The results indicate that, in the context of our SiLAN approach, selecting models that exhibit superior performance on the source test set as the source pre-trained model generally leads to optimal target adaptation outcomes.
>
> Regarding more context on the diversity of the source dataset, let us consider the synthetic (source) domain within the VisDA-2017 dataset as an illustrative example. The synthetic domain of VisDA-2017 comprises 152,397 images generated from 3D models representing 12 object categories. Each category encompasses a broad spectrum of shapes, colors, textures, and sizes, contributing to the dataset's complexity. Typically, synthetic (source) domain images exhibit clean, well-lit conditions with consistent backgrounds and lighting. Conversely, the real (target) domain images depict various backgrounds, lighting conditions, occlusions, and object poses, highlighting the diversity inherent in real-world scenarios. This diversity presents a significant challenge for domain adaptation algorithms, as they must effectively generalize across these disparities to achieve optimal performance on the target domain.
>
> We have incorporated the above discussion in a new section ***Appendix A.8 Sensitivity Analysis on The Effect of The Source Pre-Training*** in the revised manuscript and highlighted the changes in blue.

---

> > ### Author Response · Authors · 2024-03-19
> > **Response to Reviewer Bpp9 (4)**
> >
> > **Response to Broader Impact Concerns**
> >
> > We fully agree with the reviewer's suggestion that future research in SFDA should prioritize analyzing domain shifts and the biases introduced by such shifts in the source-free settings. It is essential to explore methods for interacting with potentially biased datasets, particularly in sensitive applications, within source-free settings. We sincerely appreciate the reviewer for highlighting this valuable direction for future research in source-free domain adaptation.

---

> ### Author Response · Authors · 2024-03-19
> **Response to Reviewer Bpp9 (2)**
>
> **Lack of Ablation Studies**
>
> We would like to clarify that we have conducted comprehensive ablation studies to understand the specific impact of the SiLAN augmentation and its interactions with other components of the framework. These studies are detailed in ***Appendices A.1, A.2, A.3, and A.4*** of the manuscript. If the reviewer has any specific suggestions for additional ablation experiments, we would appreciate their input and incorporate the suggested experiments in the revised manuscript.
>
> **Weakness 2 and Requested Changes 1 (Performance on ViTs)**
>
> We appreciate the reviewer's suggestion regarding ablation studies to evaluate the effectiveness of our SiLAN across different backbone architectures. Following this suggestion, we conducted experiments on all benchmark datasets using ***ViT-B-16 [2]***. Moreover, we demonstrate the continued efficacy of our proposed SiLAN in enhancing existing vision transformer-based SFDA frameworks. A new section titled ***Appendix A.9: Additional Experiments on Vision Transformers*** has been added to the revised manuscript to illustrate these findings. For reference, we have demonstrated the results in the below tables.
>
> ***Table. Comparison of the SFDA methods on Office-31 (ViT-B-16).***
>
> |  Method | A→D |A→W | D→W | D→A | W→D | W→A | Avg. |
> |:------------:|:--------:|:--------:|:--------:|:--------:|--------:|:--------:|:--------:|
> | ViT-B | 90.8 | 90.4 | 76.8 | 98.2 | 76.4 | 100.0 | 88.8 |
> | TVT [1] | 96.4 | 96.4 | 84.9 | 99.4 | 86.1 | 100.0 | 93.8 |
> | CDTrans [2] | 97.0 | 96.7 | 81.1 | 99.0 | 81.9 | 100.0 | 92.6 |
> |:------------:|:--------:|:--------:|:--------:|:--------:|--------:|:--------:|:--------:|
> |***SiLAN***| 95.3 | 97.2 | 88.1 | 99.6 | 89.2 | 100.0 | 94.6 |
> |***CDTrans+SiLAN***| 97.4 | 98.1 | 83.4 | 99.4 | 83.3 | 100.0 | 93.6 |
>
> ***Table. Comparison of the SFDA methods on Office-Home (ViT-B-16).***
>
> |  Method | Ar→Cl | Ar→Pr | Ar→Rw |Cl→Ar | Cl→Pr | Cl→Rw | Pr→Ar | Pr→Cl | Pr→Rw | Rw→Ar | Rw→Cl | Rw→Pr | Avg. |
> |:------------:|:--------:|:--------:|:--------:|:--------:|--------:|:--------:|:--------:|:--------:|:--------:|:--------:|--------:|:--------:|:--------:|
> | ViT-B | 61.8 | 79.5 | 84.3 | 75.4 | 78.8 | 81.2 | 72.8 | 55.7 | 84.4 | 78.3 | 59.3 | 86.0 | 74.8 |
> | TVT [1] | 74.9 | 86.8 | 89.5 | 82.8 | 88.0 | 88.3 | 79.8 | 71.9 | 90.1  |85.5 | 74.6 | 90.6 | 83.6 |
> | CDTrans [2] | 68.8 | 85.0 | 86.9 | 81.5 | 87.1 | 87.3 | 79.6 | 63.3 | 88.2 | 82.0 | 66.0 | 90.6 | 80.5 |
> |:------------:|:--------:|:--------:|:--------:|:--------:|--------:|:--------:|:--------:|:--------:|:--------:|:--------:|--------:|:--------:|:--------:|
> |***SiLAN***| 70.4 | 90.5 | 88.6 | 85.3 | 83.1 | 86.5 | 85.2 | 73.9 | 88.6  |86.1 | 80.0 | 92.5 | 84.2 |
> |***CDTrans+SiLAN*** | 72.1 | 86.2 | 85.4 | 81.7 | 88.5 | 86.9 | 82.4 | 81.1 | 87.9  |84.5 | 78.2 | 90.4 | 83.8 |
>
> ***Table. Comparison of the SFDA methods on VisDA-2017 (ViT-B-16).***
>
> |  Method | plane |  bcycl | bus |car | horse | knife | mcycl | person | plant | sktbrd | train | truck | Avg |
> |:------------:|:--------:|:-------:|:--------:|:-------:|:--------:|:-------:|:--------:|:-------:|:--------:|:-------:|:--------:|:-------:|-------:|
> | ViT-B | 97.7 | 48.1 | 86.6 | 61.6 | 78.1 | 63.4 | 94.7 | 10.3 | 87.7 | 47.7 | 94.4 | 35.5 | 67.1 |
> | TVT [1] |92.9 | 85.6 | 77.5 | 60.5 | 93.6 | 98.2 | 89.4 | 76.4 | 93.6 | 92.0 | 91.7 | 55.7 | 83.9 |
> | CDTrans [2] | 97.1 | 90.5 | 82.4 | 77.5 | 96.6 | 96.1 | 93.6 | 88.6 | 97.9 | 86.9 | 90.3 | 62.8 | 88.4 |
> |:------------:|:--------:|:-------:|:--------:|:-------:|:--------:|:-------:|:--------:|:-------:|:--------:|:-------:|:--------:|:-------:|-------:|
> |***SiLAN*** | 92.5 | 90.1 | 92.4 | 70.6 | 92.1 | 98.5 | 95.8 | 89.2 | 94.5 | 93.3 | 90.6 | 64.4 | 88.7 |
> |***CDTrans+SiLAN*** | 96.8 | 92.5 | 86.2 | 75.2 | 98.5 | 95.5 | 95.8 | 90.6 | 98.2 | 92.1 | 87.5 | 63.2 | 89.3 |

---

> ### Author Response · Authors · 2024-03-19
> **Response to Reviewer Bpp9 (3)**
>
> **Weakness 3 and Requested Changes 2 (Computation Overhead and Tradeoff Between Performance and Efficacy)**
>
> We would like to thank the reviewer for suggesting additional ablation studies on computational analysis to further strengthen the practical value of the method.
>
> As suggested, we conducted a runtime analysis to compare the training time for a single epoch and the overall convergence time of our SiLAN model with other SFDA methods. Specifically, we compared our results with the advanced SFDA methods AaD [3], NRC [4], and DaC [5], which served as baseline methods for our comparisons. To further investigate the tradeoff between computational overhead and factors such as dataset size and model complexity, we performed this computational analysis on both the small-scale dataset ***Office-31*** using ***ResNet-50*** and the large-scale dataset ***VisDA-2017*** using ***ResNet-101***. All experiments were conducted on a machine equipped with an Nvidia V100 GPU.
>
> Moreover, as our SiLAN can also serve as a general latent augmentation method to enhance other SFDA methods, we have included a runtime analysis for the scenario where our SiLAN latent augmentation is applied to improve the informativeness of the latent features of the neighbors for AaD [3] (referred to as ***AaD+SiLAN***). The analysis presents both the performance improvement and the extra runtime incurred compared to using AaD alone. This information equips practitioners with the necessary insights to balance performance gains with runtime considerations.
>
> The tables below illustrate that our SiLAN, NRC, and AaD exhibit similar target adaptation times per epoch, whereas DaC requires substantially more time due to its adaptive process and self-training steps, such as pseudo label generation and re-training. This observation aligns with the fact that AaD, NRC, and our SiLAN are all rooted in neighborhood searching and involve aligning predictions between query predictions and those of the neighbors. Meanwhile, our SiLAN, along with AaD and NRC, typically converge within about 10 epochs for the VisDA-2017 dataset, whereas DaC requires around 20 epochs. Notably, on the small-scale Office-31 dataset, our SiLAN achieves faster convergence compared to other methods. We hypothesize that this is because SiLAN, serving as an augmentation method, significantly enhances model convergence, particularly in situations where data is scarce. The total convergence time for target adaptation is also provided for comparison.
>
> Therefore, we conclude that despite our SiLAN having a similar one-epoch runtime compared to NRC and AaD, its use as latent augmentation leads to faster convergence compared to other SFDA methods, resulting in overall runtime benefits. A comprehensive runtime analysis is provided in ***Appendix A.6 Computational Analysis*** of our revised manuscript. Additionally, we have included the results of the computational analysis below for reference.
>
> ***Table. Time Analysis of One-Epoch Target Adaptation on Office-31 A-D (ResNet-50).***
>
> |  Method | Time (second) | One-Epoch Performance (%) |
> |:---------------:|:------:|:------:|
> | NRC [4] | 21.4 | 85.3 |
> | DaC [5] | 40.5| 80.6|
> | AaD [3] | 19.8 | 84.4 |
> |:---------------:|:------:|:------:|
> | ***SiLAN (ours)*** | 20.6 | 84.9 |
> | ***AaD+SiLAN (ours)*** | 20.2 ($+0.4$) | 85.6 ($+1.2$) |
>
> ***Table. Convergence Time Analysis for Target Adaptation on Office-31 A-D (ResNet-50).***
>
> |  Method | Time (second) | Best Performance (%)  |
> |:---------------:|:------:|:------:|
> | NRC [4] | 856.7 | 96.0 |
> | DaC [5] | 1417.5 | 94.2 |
> | AaD [3] | 714.9 | 96.4 |
> |:---------------:|:------:|:------:|
> | ***SiLAN (ours)*** | 618.9 | 97.1 |
> | ***AaD+SiLAN (ours)*** | 584.5 ($-130.4$) | 97.5 ($+1.1$) |
>
> ***Table. Time Analysis of One-Epoch Target Adaptation on VisDA-2017 Dataset (ResNet-101).***
>
> |  Method | Time (second) | One-Epoch Performance (%) |
> |:---------------:|:------:|:------:|
> | NRC [4] | 469.2 | 79.2 |
> | DaC [5] | 632.8 | 82.4 |
> | AaD [3] | 453.6 | 82.6 |
> |:---------------:|:------:|:------:|
> | ***SiLAN (ours)*** | 465.9 | 84.5 |
> | ***AaD+SiLAN (ours)*** | 462.3 ($+8.7$) | 86.8 ($+4.2$) |
>
> ***Table. Convergence Time Analysis for Target Adaptation on VisDA-2017 Dataset (ResNet-101).***
>
> |  Method | Time (second) | Best Performance (%) |
> |:---------------:|:------:|:------:|
> | NRC [4] | 4692.8 | 85.9 |
> | DaC [5] | 12656.3 | 87.3 |
> | AaD [3] | 4536.2 | 87.1 |
> |:---------------:|:------:|:------:|
> | ***SiLAN (ours)*** | 3727.2 | 88.3 |
> | ***AaD+SiLAN (ours)*** | 3236.1 ($-1300.1$)  | 89.7 ($+2.6$) |

---

> ### Author Response · Authors · 2024-03-19
> **Reference**
>
> [1] Jinyu Yang, Jingjing Liu, Ning Xu, and Junzhou Huang. Tvt: Transferable vision transformer for unsu- pervised domain adaptation. In Proceedings of the IEEE/CVF Winter Conference on Applications of Computer Vision (CVPR), pp. 520–530, 2023.
>
> [2] Tongkun Xu, Weihua Chen, Pichao Wang, Fan Wang, Hao Li, and Rong Jin. Cdtrans: Cross-domain transformer for unsupervised domain adaptation. arXiv preprint arXiv:2109.06165, 2021
>
> [3] Shiqi Yang, Yaxing Wang, Kai Wang, Shangling Jui, et al. Attracting and dispersing: A simple approach for source-free domain adaptation. In Conference on Neural Information Processing Systems (NeurIPS), 2022.
>
> [4] Shiqi Yang, Joost van de Weijer, Luis Herranz, Shangling Jui, et al. Exploiting the intrinsic neighborhood structure for source-free domain adaptation. Advances in Neural Information Processing Systems, 34: 29393–29405, 2021a.
>
> [5] Ziyi Zhang, Weikai Chen, Hui Cheng, Zhen Li, Siyuan Li, Liang Lin, and Guanbin Li. Divide and contrast: Source-free domain adaptation via adaptive contrastive learning.
> In Conference on Neural Information Processing Systems (NeurIPS), 2022.

---

> ### Author Response · Authors · 2024-04-03
> **Prompt for Discussion: A Friendly Reminder**
>
> Dear Reviewer Bpp9,
>
> We extend our sincere gratitude for your invaluable suggestions regarding our paper.
>
> We believe we have adequately addressed all your concerns in the responses provided above. However, should you have any further inquiries or require additional clarification on our responses, we are more than willing to engage in further discussion.
>
> Once again, we greatly appreciate your diligent efforts in reviewing our paper.
>
> Best regards,
>
> Authors 2099

---

### Review · Reviewer_KBch · 2024-03-05

**Summary Of Contributions:**

This paper presents a theoretical analysis of contrastive learning for source-free domain adaptation (SFDA), identifying key factors that influence its effectiveness. Motivated by the findings, the authors propose a novel latent augmentation method called Source-informed Latent Augmented Neighborhood (SiLAN), which leverages the dispersion of latent features guided by the source pre-trained model to reduce logit group overlaps and enhance positive key generation. SiLAN is integrated into the InfoNCE contrastive learning framework and achieves state-of-the-art performance on multiple SFDA benchmark datasets.

**Audience:**

Yes

**Claims And Evidence:**

Yes

**Requested Changes:**

While the theoretical analysis provides valuable insights, some parts (e.g., Proposition 3 and Lemma 4) are quite technical and may be difficult for some readers to fully grasp. It would be helpful if the authors could provide more intuition or visual illustrations to make these aspects more accessible.

I also have some questions:

1. How sensitive is SiLAN's performance to the choice of hyperparameters (K, τ, etc.)? Can the authors provide some general guidelines for setting these values?

2. The theoretical analysis assumes convergence of the contrastive loss. How well does this assumption hold in practice, and how might non-convergence impact the performance of SiLAN?

3. Could the proposed latent augmentation be combined with other advanced techniques (e.g., adversarial learning, pseudo-labeling) to further boost SFDA performance?

4. The method is evaluated on image classification tasks. How might it generalize to other data modalities or tasks (e.g., semantic segmentation, object detection)?

**Strengths And Weaknesses:**

## Strengths

1. The paper provides a rigorous theoretical analysis of contrastive SFDA, offering valuable insights into the key factors influencing its performance. The analysis is well-motivated by empirical observations of target feature dispersion and neighborhood informativeness, and clearly presented.

2. The proposed SiLAN method is a simple yet effective approach that addresses several limitations of existing contrastive SFDA methods identified in the theoretical analysis. It is grounded in the theoretical findings and leverages both neighborhood exploration and source model knowledge.

3. SiLAN is evaluated on multiple benchmark datasets, with results consistently showing it outperforming state-of-the-art SFDA methods. Ablation studies are also provided to justify the design choices.

4. The paper advances both the theoretical understanding and practical methodology of contrastive learning for the challenging problem of source-free domain adaptation. It is well-structured and clearly written, making it easy to follow the main ideas and contributions.

## Weaknesses

1. The hyperparameter selection for SiLAN (e.g., number of neighbors K, temperature τ) is not extensively discussed. More guidance or sensitivity analysis on these choices would be helpful for practitioners. How sensitive is SiLAN's performance to these hyperparameters, and can the authors provide general guidelines for setting their values?

2. The computational cost of SiLAN compared to other SFDA methods is not reported. An analysis of the tradeoff between performance gain and additional computation would further strengthen the practical value of the method. What is the computational overhead of SiLAN, and how does it scale with the size of the dataset or model?

---

> ### Author Response · Authors · 2024-03-19
> **Author Response to Reviewer KBch (1)**
>
> We hugely thank the reviewer for acknowledging our work and providing valuable suggestions to improve our paper. In our response, we will address the identified weaknesses and incorporate the requested changes systematically, as outlined below:
>
> **Weakness 1 and Question 1 (More Guidance for Hyperparameter Selection)**
>
> We appreciate the reviewer's suggestion to provide additional guidance for practitioners in selecting hyperparameters for our SiLAN model.
>
> As far as we are concerned, our SiLAN framework introduces three additional hyperparameters: the number of the source-informed kNNs $K_s$, the number of target kNNs $K_t$, and the logit temperature for contrastive loss \tau. In the original submission of our paper, a detailed sensitivity analysis of these hyperparameters, including $K_s$, $K_t$, and $\tau$ on the Office-31 dataset, was presented in ***Appendices A.1, A.2, and A.3***.
>
> In response to the reviewer's suggestion, we have added a new section to the revised manuscript. This section offers general guidance for selecting hyperparameters, drawing insights from the experimental results of the sensitivity analysis presented in ***Appendices A.1, A.2, and A.3***. Our aim with this added discussion is to offer clearer guidance for practitioners on hyperparameter selection when utilizing our SiLAN. The revised sections have been highlighted in blue for easy identification.
>
> For the reviewer's reference, we have included the added section ***Appendix A.5*** on hyperparameter selection guidance below:
>
> ***Appendix A.5: General Guidance for Hyperparameter Selection***
>
> Our InfoNCE-based SiLAN introduces three additional hyperparameters: the number of the source-informed kNNs $K_s$, the number of target kNNs $K_t$, and the logit temperature for contrastive loss $\tau$. In this section, we will offer general guidance for setting values for these hyperparameters based on the results of the sensitivity analysis obtained from Appendices A.1, A.2, and A.3.
>
> To identify the optimal set of hyperparameters, we recommend the following systematic approach: begin by determining the number of target $k$NNs, denoted as $K_t$. This parameter determines the mean of the Gaussian for our latent augmentation, directly influencing the effectiveness of contrastive clustering. In general, $K_t$ can be roughly estimated based on the number of samples per class in the target dataset; a smaller target dataset should use a smaller $K_t$. For instance, our experiments show that the optimal $K_t$ ranges from 3 to 5 for Office datasets, each containing about 200 images per class. However, for VisDA-2017, which comprises around 23K images per class, the optimal $K_t$ is 15. Therefore, to determine the optimal $K_t$, we suggest a search range from 2 to 8 for a target dataset with 100 to 1K samples per class, and a search range from 10 to 20 for a target dataset with more than 20K images per class.
>
> Subsequently, the number of the source-informed $k$NNs, denoted as $K_s$, which determines the standard deviation of the Gaussian for our SiLAN, could be decided based on $K_t$. Our analysis in Section 5 indicates that $K_s$ should be large enough to contain the farthest source-informed neighbor that may share the same ground truth as the target query. However, it should not be excessively large to avoid including features with inconsistent ground truth in the positive key generation. We found that the standard deviation of these source-informed kNNs' features is a reliable indicator for selecting $K_s$ to define the optimal latent augmentation region. As detailed in Appendix A.2, a larger standard deviation of the latent feature vectors of the source-informed kNNs generally correlates with better performance on target classification tasks. Typically, the largest standard deviation occurs when $K_s$ is approximately equal to $K_t$. Therefore, a preliminary strategy could be setting $K_s$ equal to $K_t$, with the optimal $K_s$ lying within the range of $K_t \pm 2$.
>
> Finally, the temperature $\tau$ for contrastive logits should be determined similarly to self-supervised learning frameworks, as regardless of the mathematical space in which clustering occurs, this parameter influences the degree of penalization applied to hard negative samples (Wang et al., 2021). Unlike unsupervised representation learning, which aims for a universal representation across tasks, our contrastive objective operates in the output logit space, favoring a small value for $\tau$ (similar to identifying the temperature in knowledge distillation frameworks; Hinton et al., 2015). In our experiments, the optimal value for $\tau$ falls between 0.07 and 0.15. Therefore, a search range between 0.05 and 0.2 is recommended for practitioners.

---

> ### Author Response · Authors · 2024-03-19
> **Author Response to Reviewer KBch (2)**
>
> **Weakness 2 (Computational Analysis and Tradeoff)**
>
> ***Overall Computational Analysis***
>
> We would like to thank the reviewer for suggesting additional ablation studies on computational analysis to further strengthen the practical value of the method.
>
> Following the suggestion, we conducted a runtime analysis to compare the training time for a single epoch and the overall convergence time of our SiLAN model with other SFDA methods. Specifically, we compared our results with the advanced SFDA methods AaD [1], NRC [5], and DaC [6], which served as baseline methods for our comparisons.
>
> ***Computational Analysis Scaling with Dataset and Model Size***
>
> Furthermore, in response to the reviewer's suggestion to demonstrate scalability with respect to dataset size and model complexity, we performed this computational analysis on both the small-scale dataset ***Office-31*** using ***ResNet-50*** and the large-scale dataset ***VisDA-2017*** using ***ResNet-101***. All experiments were conducted on a machine equipped with an Nvidia V100 GPU.
>
> ***Analysis of Tradeoffs When Combining with Other Methods***
>
> Moreover, as our SiLAN can also serve as a general latent augmentation method to enhance other SFDA methods, we have included a runtime analysis for the scenario where our SiLAN latent augmentation is applied to improve the informativeness of the latent features of the neighbors for AaD [1] (referred to as "***AaD+SiLAN***"). In this analysis, we present the performance gain and additional runtime compared to AaD alone, providing practitioners with the information needed to balance performance gains against runtime considerations.
>
> ***Experimental Results of Computational Analysis***
>
> The tables below illustrate that our SiLAN, NRC, and AaD exhibit similar target adaptation times per epoch, whereas DaC requires substantially more time due to its adaptive process and self-training steps, such as pseudo label generation and re-training. This observation aligns with the fact that AaD, NRC, and our SiLAN are all rooted in neighborhood searching and involve aligning predictions between query predictions and those of the neighbors. Meanwhile, our SiLAN, along with AaD and NRC, typically converge within about 10 epochs for the VisDA-2017 dataset, whereas DaC requires around 20 epochs. Notably, on the small-scale Office-31 dataset, our SiLAN achieves faster convergence compared to other methods. We hypothesize that this is because SiLAN, serving as an augmentation method, significantly enhances model convergence, particularly in situations where data is scarce. The total convergence time for target adaptation is also provided for comparison.
>
> Therefore, we conclude that despite our SiLAN having a similar one-epoch runtime compared to NRC and AaD, its use as latent augmentation leads to faster convergence compared to other SFDA methods, resulting in overall runtime benefits. A comprehensive runtime analysis is provided in ***Appendix A.6*** of our revised manuscript. Additionally, we have included the results of the computational analysis below for reference.
>
> ***Table. Time Analysis of One-Epoch Target Adaptation on Office-31 A-D (ResNet-50).***
>
> |  Method | Time (second) | One-Epoch Performance (%) |
> |:---------------:|:------:|:------:|
> | NRC [5] | 21.4 | 85.3 |
> | DaC [6] | 40.5| 80.6|
> | AaD [1] | 19.8 | 84.4 |
> |:---------------:|:------:|:------:|
> | ***SiLAN (ours)*** | 20.6 | 84.9 |
> | ***AaD+SiLAN (ours)*** | 20.2 ($+0.4$) | 85.6 ($+1.2$) |
>
> ***Table. Convergence Time Analysis for Target Adaptation on Office-31 A-D (ResNet-50).***
>
> |  Method | Time (second) | Best Performance (%)  |
> |:---------------:|:------:|:------:|
> | NRC [5] | 856.7 | 96.0 |
> | DaC [6] | 1417.5 | 94.2 |
> | AaD [1] | 714.9 | 96.4 |
> |:---------------:|:------:|:------:|
> | ***SiLAN (ours)*** | 618.9 | 97.1 |
> | ***AaD+SiLAN (ours)*** | 584.5 ($-130.4$) | 97.5 ($+1.1$) |
>
> ***Table. Time Analysis of One-Epoch Target Adaptation on VisDA-2017 Dataset (ResNet-101).***
>
> |  Method | Time (second) | One-Epoch Performance (%) |
> |:---------------:|:------:|:------:|
> | NRC [5] | 469.2 | 79.2 |
> | DaC [6] | 632.8 | 82.4 |
> | AaD [1] | 453.6 | 82.6 |
> |:---------------:|:------:|:------:|
> | ***SiLAN (ours)***| 465.9 | 84.5 |
> | ***AaD+SiLAN (ours)*** | 462.3 ($+8.7$) | 86.8 ($+4.2$) |
>
> ***Table. Convergence Time Analysis for Target Adaptation on VisDA-2017 Dataset (ResNet-101).***
>
> |  Method | Time (second) | Best Performance (%) |
> |:---------------:|:------:|:------:|
> | NRC [5] | 4692.8 | 85.9 |
> | DaC [6] | 12656.3 | 87.3 |
> | AaD [1] | 4536.2 | 87.1 |
> |:---------------:|:------:|:------:|
> | ***SiLAN (ours)*** | 3727.2 | 88.3 |
> | ***AaD+SiLAN (ours)*** | 3236.1 ($-1300.1$)  | 89.7 ($+2.6$) |

---

> ### Author Response · Authors · 2024-03-19
> **Author Response to Reviewer KBch (3)**
>
> **Requested Changes (Enhanced Intuitive Understanding and Visual Representation of Theoretical Statements)**
>
> We appreciate the reviewer's suggestion to enhance the clarity of certain technique propositions and lemmas (e.g., Proposition 3 and Lemma 4) by providing more intuition and contextual knowledge. We have carefully revised the manuscript to include more intuitive explanations and illustrations. Our goal is to ensure that readers from diverse backgrounds can grasp the technical content by providing sufficient context and details. We have marked the revisions in blue within the revised manuscript to facilitate easy identification.
>
> Due to space constraints in the main text, we have included a new section, ***Appendix C: Further Intuitive Insights into The Theoretical Work***, in the revised manuscript. This section offers additional intuitions and visual illustrations to provide practitioners with more context, aiding in their understanding of the propositions and lemmas.
>
> We have provided a concise summary of the intuitions added in the new section below for reference. For a comprehensive understanding of these intuitions and visual illustrations, we kindly request the reviewer to refer to ***Appendix C: Further Intuitive Insights into The Theoretical Work*** of the revised manuscript.
>
> ***Intuition of Proposition 3***
>
> Proposition 3 establishes a theoretical lower bound for the L1 distance between the latent features of data samples belonging to different logit clusters. It builds upon the scenario where the classification model is optimized using a contrastive loss to align query predictions with SiLAN augmentations while deliberately misaligning predictions within the same mini-batch.
>
> Our SiLAN augmentation introduces Gaussian noise, incorporating guidance from the source pre-trained model's $k$NNs to the centroid of the query neighborhood found by the current target model. This results in a Gaussian profile for the positive augmentation, enabling the establishment of a lower bound on the distance between logit clusters. To aid comprehension, we offer a visual representation of the impact of the Gaussian profile introduced by our SiLAN augmentation, as depicted in ***Figure 6*** of the revised manuscript. The noise for the augmentation (the extraneous noise introduced in Proposition 3) is drawn from a Gaussian distribution determined by the variance of the $k$NNs as determined by the source pre-trained model. Proposition 3 utilizes a geometric method inspired by computational optics to determine the effective region of a Gaussian beam, using the centroid of the target neighbors of the query as its center.
>
> ***Intuition of Lemma 4***
>
> Proposition 3 establishes the theoretical lower bound of the L1 distance between latent features belonging to different logit clusters. However, our primary interest lies not in the cluster distance within the latent feature space, but rather in the distance lower bound between the output logits of different clusters. A larger distance lower bound in the output logit space facilitates the derivation of decision boundaries for target classification.
>
> The distance lower bound in the latent feature space, as shown in Proposition 3, can be straightforwardly extended to the output logit space if the classifier is linear. Hence, we employ a linear fully-connected classifier in all our experiments to ensure consistency between our theoretical framework and empirical implementations.

---

> ### Author Response · Authors · 2024-03-19
> **Author Response to Reviewer KBch (4)**
>
> **Question 2 (Convergence of Contrastive Learning Objective)**
>
> We would like to thank the reviewer for bringing up questions regarding the guarantee of convergence of the contrastive learning objective and how non-convergence would impact our proposed SiLAN.
>
> As far as our knowledge extends, there is no such definitive theoretical proof establishing the convergence of contrastive learning objectives within the research community. Progress has been made in understanding the behaviors of contrastive clustering in specific settings and under certain assumptions, as reflected in recent works [10] and [11], akin to what we have explored in our current study in the source-free domain adaptation settings. However, despite the lack of a formal theoretical convergence proof, empirical demonstrations of the effectiveness and convergence of contrastive learning objectives abound across various tasks, including self-supervised learning [12], supervised learning [13], domain adaptation [1], and reinforcement learning [14], among others. Additionally, empirical validation of model convergence has been demonstrated across diverse data modalities, including images [12], language [16], graphs [15], and speech [14].
>
> In our experiments across all datasets, SiLAN demonstrates empirical convergence, as evidenced in the runtime analysis, requiring fewer training epochs compared to other advanced SFDA methods. We posit that SiLAN's augmentation mechanism significantly enhances model convergence, particularly in scenarios where data is limited.
>
> To empirically evaluate the impact of non-convergence on performance, we conducted an ablation study within the runtime analysis (refer to our response to ***Weakness 2 (Computational Analysis and Tradeoff)***). Specifically, we evaluated target classification performance with only one epoch adaptation (which demonstrates the case of non-convergence). The results, outlined in the runtime analysis (***Tables 6 and 8*** in ***Appendix A.6 Computational Analysis***), indicate that despite the contrastive learning objective's non-convergence, SiLAN still achieves strong performance in target-domain classification (e.g., 84.9% on Office-31 AD adaptation and 84.5% on the VisDA-2017 dataset). This superiority is particularly pronounced when compared to other advanced SFDA methods.

---

> ### Author Response · Authors · 2024-03-19
> **Author Response to Reviewer KBch (5)**
>
> **Question 3 (Integration of SiLAN with Other SFDA Techniques)**
>
> We appreciate the reviewer's suggestion to integrate additional methods to enhance the performance of our SiLAN. In our original manuscript, our focus was on maintaining a simple framework to primarily validate the proposed latent augmentation within the InfoNCE-based contrastive learning framework.
>
> However, in response to the reviewer's suggestion, we conducted additional experiments on the large-scale VisDA-2017 dataset using ResNet-101 to explore the potential of applying SiLAN as a general latent augmentation method to enhance existing SFDA methods. To be specific, we integrated our proposed SiLAN method into four SFDA frameworks: AaD [1], HCL [3], NRC [5], and A$^2$Net [7].
>
> Among them, as suggested by the reviewer, HCL is a contrastive SFDA framework built upon pseudo-labeling, and A$^2$Net is an adversarial SFDA framework. Additionally, AaD features a more advanced contrastive learning objective tailored for solving SFDA problems, while NRC utilizes a hierarchical neighborhood searching strategy. The experiments corresponding to these frameworks have been included in ***Appendix A.7: SiLAN as A General Latent Augmentation Method*** of the revised manuscript.
>
> The results of these experiments are provided below for reference. Please note that ***XXX-SiLAN*** indicates the utilization of our SiLAN method to augment the latent features of the respective SFDA framework. As shown in the table below, our proposed SiLAN enhances the performance of HCL, A$^2$Net, NRC, and AaD by 0.6%, 2.0%, 1.2%, and 2.6%, respectively. These results demonstrate that SiLAN is an effective latent augmentation method for addressing SFDA problems across various SFDA frameworks.
>
> ***Table. Ablation studies of integrating SiLAN into various SFDA frameworks for enhanced performance on VisDA2017 (ResNet-101)***.
>
> |  Method | plane |  bcycl | bus |car | horse | knife | mcycl | person | plant | sktbrd | train | truck | Avg |
> |:------------:|:--------:|:-------:|:--------:|:-------:|:--------:|:-------:|:--------:|:-------:|:--------:|:-------:|:--------:|:-------:|-------:|
> | ResNet-101 | 55.1 | 53.3 | 61.9 | 59.1 | 80.6 | 17.9 | 79.7 | 31.2 | 81.0 | 26.5 | 73.5 | 8.5 | 52.4 |
> | HCL [3] | 93.3 | 85.4 | 80.7 | 68.5 | 91.0 | 88.1 | 86.0 | 78.6 | 86.6 | 88.8 | 80.0 | 74.7 | 83.5 |
> |A$^{2}$Net [7] | 94.0 | 87.8 | 85.6 | 66.8 | 93.7 | 95.1 | 85.8 | 81.2 | 91.6 | 88.2 | 86.5 | 56.0 | 84.3 |
> | AaD [1] | 95.2 | 90.5 | 85.5 | 79.2 | 96.4 | 96.2 | 88.8 | 80.4 | 93.9 | 91.8 | 91.1 | 55.9 | 87.1|
> | NRC [5] | 96.1 | 90.8 | 83.9 | 61.5 | 95.7 | 95.7 | 84.4 | 80.7 | 94.0 | 91.9 | 89.0 | 59.5 | 85.3 |
> |:------------:|:--------:|:-------:|:--------:|:-------:|:--------:|:-------:|:--------:|:-------:|:--------:|:-------:|:--------:|:-------:|-------:|
> |***SiLAN*** | 97.5 | 90.1 | 85.8 | 80.4 | 97.6 | 95.5 | 92.0 | 82.9 | 96.5 | 95.3 | 92.6 | 53.4 | 88.3 |
> |***HCL+SiLAN*** |  94.6 | 85.4 | 83.2 | 67.3 | 94.2 | 86.5 | 86.3 | 80.8 | 88.2 | 85.2 | 83.4 | 74.6 | 84.1 |
> |***A$^{2}$Net+SiLAN*** | 97.2 | 91.2 | 87.4 | 66.8 | 96.9 | 96.9 | 88.0 | 80.9 | 93.5 | 93.3 | 91.1 | 53.2 | 86.3 |
> |***NRC+SiLAN*** | 97.2 | 90.5 | 84.3 | 63.8 | 96.7 | 95.4 | 86.2 | 85.1 | 95.6 | 93.2 | 90.2 | 59.8 | 86.5 |
> |***AaD+SiLAN*** | 98.4 |91.8 | 86.2 | 80.6 | 96.3 | 95.7 | 94.4 | 87.5 | 95.8 | 94.2 | 93.4 | 62.1 | 89.7 |
>
> **Question 4 (How Might SiLAN Generalize to Other Data Modalities or Tasks)**
>
> We would like to thank the reviewer for raising the question. Our proposed SiLAN augmentation technique, which operates in the latent space (the output space of the encoder) within the SFDA framework, is designed to be applicable across various tasks and data modalities with minimal modification efforts. Additionally, the informativeness of SiLAN stems from the feature clusters formed by the $k$NN algorithm, which still remains independent of specific data modalities and underlying tasks, provided that the encoder and task-specific head are appropriately designed.
>
> Therefore, to handle various data modalities and tasks, practitioners can integrate our proposed SiLAN within the latent space between their encoder and task-specific head. For example, SiLAN could be applied to the output vector of SegNet's encoder for semantic segmentation tasks.

---

> ### Author Response · Authors · 2024-03-19
> **Reference**
>
> [1] Shiqi Yang, Yaxing Wang, Kai Wang, Shangling Jui, et al. Attracting and dispersing: A simple approach for source-free domain adaptation. In Conference on Neural Information Processing Systems (NeurIPS), 2022.
>
> [2] Jian Liang, Dapeng Hu, and Jiashi Feng. Do we really need to access the source data? source hypothesis transfer for unsupervised domain adaptation. In International Conference on Machine Learning (ICML), pp. 6028–6039. PMLR, 2020.
>
> [3] Jiaxing Huang, Dayan Guan, Aoran Xiao, and Shijian Lu. Model adaptation: Historical contrastive learning for unsupervised domain adaptation without source data.
> Advances in Neural Information Processing Systems (NeurIPS), 34:3635–3649, 2021a.
>
> [4] Shiqi Yang, Yaxing Wang, Joost van de Weijer, Luis Herranz, and Shangling Jui.
> Generalized source-free domain adaptation. In Proceedings of the IEEE/CVF International Conference on Computer Vision (ICCV), pp. 8978–8987, 2021b.
>
> [5] Shiqi Yang, Joost van de Weijer, Luis Herranz, Shangling Jui, et al. Exploiting the intrinsic neighborhood structure for source-free domain adaptation. Advances in Neural Information Processing Systems, 34: 29393–29405, 2021a.
>
> [6] Ziyi Zhang, Weikai Chen, Hui Cheng, Zhen Li, Siyuan Li, Liang Lin, and Guanbin Li. Divide and contrast: Source-free domain adaptation via adaptive contrastive learning.
> In Conference on Neural Information Processing Systems (NeurIPS), 2022.
>
> [7] Xia, H., Zhao, H. and Ding, Z. Adaptive adversarial network for source-free domain adaptation. In Proceedings of the IEEE/CVF international conference on computer vision (CVPR), 2021.
>
> [8] Yang Liu, Wei Zhang, and June Wang, Source-free domain adaptation for semantic segmentation. In Proceedings of the IEEE/CVF Conference on Computer Vision and Pattern Recognition (CVPR), 2021.
>
> [9] Uiwon Hwang, Jonghyun Lee, Juhyeon Shin, and Sungroh Yoon. Sf(da)^2: Source-free domain adaptation through the lens of data augmentation. In The Twelfth International Conference on Learning Representations (ICLR), 2024.
>
> [10] Feng Wang and Huaping Liu. Understanding the behaviour of contrastive loss. In Proceedings of the IEEE/CVF Conference on Computer Vision and Pattern Recognition (CVPR), pp. 2495–2504, 2021.
>
> [11] Weiran Huang, Mingyang Yi, and Xuyang Zhao. Towards the generalization of contrastive self-supervised learning. arXiv preprint arXiv:2111.00743, 2021b.
>
> [12] Ting Chen, Simon Kornblith, Mohammad Norouzi, and Geoffrey Hinton. A simple framework for contrastive learning of visual representations. In International Conference on Machine Learning (ICML), pp. 1597– 1607. PMLR, 2020.
>
> [13] Prannay Khosla, Piotr Teterwak, Chen Wang, Aaron Sarna, Yonglong Tian, Phillip Isola, Aaron Maschinot, Ce Liu, and Dilip Krishnan. Supervised contrastive learning. Advances in Neural Information Processing Systems (NeurIPS), 33:18661–18673, 2020.
>
> [14] Aaron van den Oord, Yazhe Li, and Oriol Vinyals. Representation learning with contrastive predictive coding. arXiv preprint arXiv:1807.03748, 2018.
>
> [15] Yuning You, Tianlong Chen, Yongduo Sui, Ting Chen, Zhangyang Wang, and Yang Shen. Graph contrastive learning with augmentations. Advances in neural information processing systems (NeurIPS), 33:5812–5823, 2020.
>
> [16] Jinyu Yang, Jiali Duan, Son Tran, Yi Xu, Sampath Chanda, Liqun Chen, Belinda Zeng, Trishul Chilimbi, and Junzhou Huang. Vision-language pre-training with triple contrastive learning. In Proceedings of the IEEE/CVF Conference on Computer Vision and Pattern Recognition (CVPR), pp. 15671–15680, 2022a

---

> > ### Comment · Reviewer_KBch · 2024-03-24
> >
> > Thank you a lot for the detailed responses and refinement of the manuscript. My concerns are addressed.

---

> > > ### Author Response · Authors · 2024-03-25
> > > **Huge Thanks to Reviewer KBch**
> > >
> > > Huge thanks for your quick response and the meticulous effort you dedicated to reviewing our manuscript. Your invaluable suggestions have significantly contributed to refining our paper, and we deeply appreciate the guidance and expertise you have shared with us.

---

### Author Response · Authors · 2024-04-05
**Huge Thanks for the Dedicated Effort and Priceless Contributions of the Area Chair and Reviewers!**

Dear Area Chair and Reviewers,

We extend our deepest gratitude for your dedication and huge effort throughout the review process, as well as your invaluable suggestions that have significantly enhanced our work and expanded its potential impact.

We are committed to meticulously addressing and incorporating responses to each comment provided by the AC and reviewers in our camera-ready submission.

Once again, we express our profound appreciation to all who have contributed their expertise to refine our paper. It is a tremendous honor to receive such insightful guidance from esteemed experts in this field.

Best regards,

The Authors of Manuscript 2099